# PROGRAM: PROtotype GRAph Model based Pseudo-Label Learning for Test-Time Adaptation

**Haopeng Sun**[2*]    **Lumin Xu**[3]    **Sheng Jin**[4,2]    **Ping Luo**[4,5]    **Chen Qian**[1,2 ✉]    **Wentao Liu**[2]

[1] Department of Computer Science and Technology, Tsinghua University, Beijing, China
[2] SenseTime Research and Tetras.AI    [3] The Chinese University of Hong Kong
[4] The University of Hong Kong    [5] Shanghai AI Laboratory
{sunhaopeng,jinsheng}@tetras.ai   qianc18@mails.tsinghua.edu.cn

## ABSTRACT

Test-time adaptation (TTA) aims to adapt a pre-trained model from a source domain to a target domain only using online unlabeled target data during testing, without accessing to the source data or modifying the original training process. Among the various TTA methods, pseudo-labeling has gained popularity. However, the presence of incorrect pseudo-labels can hinder the effectiveness of target domain adaptation. To overcome this challenge, we propose a novel TTA method, called **PRO**totype **GRA**ph **M**odel based pseudo-label learning (**PROGRAM**). PROGRAM consists of two key components: (1) Prototype Graph Model (PGM) for reliable pseudo-label generation; (2) Robust Self-Training (RST) for test-time adaptation with noisy pseudo-labels. PGM constructs the graph using prototypes and test samples, facilitating effective message passing among them to generate more reliable pseudo-labels. RST combines the advantages of consistency regularization and pseudo-labeling to achieve robust target domain adaptation in the presence of noisy pseudo-labels. Our proposed PROGRAM can be easily integrated into existing baselines, resulting in consistent improvement. Extensive experiments show that our PROGRAM outperforms the existing TTA methods on multiple domain generalization and image corruption benchmarks.

## 1 INTRODUCTION

Deep neural network (DNN) based methods perform exceedingly well when training and testing data are sampled from the same distribution. However, under test-time domain shift (Pan & Yang, 2009; Gopalan et al., 2011), DNNs encounter significant performance degradation. Test-time adaptation (TTA) (Liang et al., 2023) is a prominent paradigm to alleviate this problem. TTA methods adapt the models trained from the source domain to the target domain at test-time, without accessing to labeled data from the target domain.

A majority of TTA approaches adopt pseudo-labeling (PL) based methods, which first generate pseudo-labels and then adapt the model to the target domain via self-training. Traditional PL based methods (Wang et al., 2021a; Lee et al., 2013) directly use the output of the classifier as pseudo-labels, and reinforce the model to learn overly confident predictions for the test data. Due to inevitable noises contained in the pseudo-labels, such approaches might suffer from dramatic performance degradation (Mukhoti et al., 2020) or total model collapse (Wang et al., 2022b). In order to generate more reliable pseudo-labels, we propose a novel TTA method, termed **PRO**totype **GRA**ph **M**odel based pseudo-label learning (**PROGRAM**).

In order to generate credible pseudo-labels, prototype-based PL (Iwasawa & Matsuo, 2021; Wang et al., 2023; Liang et al., 2020), and nearest-neighbor based PL (Yang et al., 2021; Jang et al., 2023) have been proposed for performance improvement. However, these two types of approaches have

---

\* Work done during internship at SenseTime Research and Tetras.AI.
✉ Corresponding author.

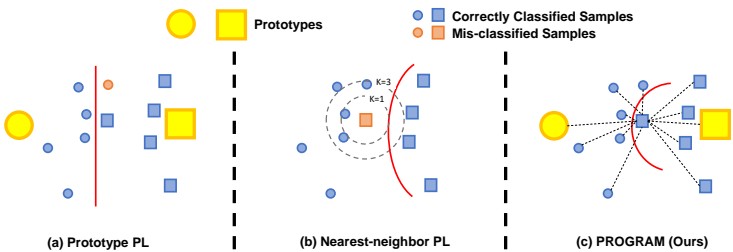

Figure 1: Different strategies of pseudo-labeling (PL): (a) Prototype-based PL. (b) Nearest-neighbor based PL. (c) PROtotype GRAph Model based pseudo-label learning (PROGRAM). The red lines represent the decision boundary.

their own drawbacks. As shown in Fig. 1(a), prototype-based PL generates pseudo-labels based on *global* class prototype information but fails to incorporate local features of test samples, leading to incorrect predictions under domain shift. As shown in Fig. 1(b), nearest-neighbor based PL relies on *local* neighboring features of test samples for pseudo-label generation, overlooking the global representative features of each class. As the feature distribution of different categories may have overlaps under domain shift, test samples near the decision boundary are easily influenced by neighbors from other categories. To mitigate these problems, we propose Prototype Graph Model (PGM) to take both global and local information into consideration for pseudo-label generation. As shown in Fig. 1(c), our proposed method combines the advantages of both prototype-based PL and nearest-neighbor based PL, by incorporating both the global representative features of prototypes and the local information from neighboring test data in a flexible graph representation.

In terms of the self-training stage, some approaches (Lee et al., 2013; Iwasawa & Matsuo, 2021; Jang et al., 2023) use *hard* pseudo-labels (*i.e.* one-hot encoding), while others (Wang et al., 2023) use *soft* pseudo-labels (*i.e.* probability encoding) for model fine-tuning. Hard labels help accelerate model training but are vulnerable to label noises, so it is generally necessary to set a handpicked threshold (Lee et al., 2013) to filter out unreliable pseudo-labels (Rizve et al., 2021). However, choosing a universally appropriate threshold is challenging for different datasets (Wang et al., 2022a). Too high threshold will discard some useful correct pseudo-labels, while too low threshold will involve noisy pseudo-labels. On the contrary, soft labels help improve the model generalization ability, but suffer from slower model convergence. In this paper, we propose Robust Self-Training (RST) technique to avoid the above pitfalls. When the pseudo-labels are reliable enough (the model predictions are consistent with pseudo-labels), we adopt hard pseudo-labels to accelerate model convergence without setting handpicked thresholds. Otherwise, we adopt soft pseudo-labels and use noise-resistant consistency regularization losses for model training. Our proposed RST effectively combines consistency regularization and pseudo-labeling, without tediously setting handcrafted threshold.

We benchmark PROGRAM on four domain generalization datasets (*i.e.*, VLCS (Torralba & Efros, 2011), PACS (Li et al., 2017), OfficeHome (Venkateswara et al., 2017) and TerraIncognita (Beery et al., 2018)) and three image corruption benchmarks (Hendrycks & Dietterich, 2019) including CIFAR-10C, CIFAR-100C, and ImageNet-C. Experiments show that PROGRAM outperforms other TTA methods and can be applied to various baselines to deliver consistent improvements.

Our main contributions can be summarized as follows:

- We propose a novel TTA method called PROtotype GRAph Model based pseudo-label learning (PROGRAM) to address the issue of noisy pseudo-labels. Extensive experiments on popular domain generalization and image corruption benchmarks demonstrate the superiority of PROGRAM over the previous state-of-the-art TTA methods.

- In the pseudo-label generation stage, we propose Prototype Graph Model (PGM) that combines the advantages of both prototype-based PL and nearest-neighbor based PL to produce reliable pseudo-labels. It incorporates both global representative features of prototypes and local information from neighboring test data in a flexible graph representation.

- In the self-training stage, we propose Robust Self-Training (RST) that combines pseudo-labeling and consistency regularization. It benefits from both hard and soft pseudo-labels to make the self-training process more stable and robust.

## 2 RELATED WORK

### 2.1 TEST-TIME ADAPTATION

Test-time adaptation (TTA) (Liang et al., 2023) involves adapting a pre-trained model using online unlabeled test data only. It is closely related to two research topics, *i.e.* test-time training (TTT) and source-free domain adaptation (SFDA). TTT methods (Gidaris et al., 2018; Liu et al., 2021; Sun et al., 2020b) also optimize during testing, however, they require to alter training procedure by introducing the proxy loss on source data. SFDA methods (You et al., 2021b; Wang et al., 2021b; Yan et al., 2021; Liang et al., 2020; Morerio et al., 2020; Kurmi et al., 2021; Zhou et al., 2022; Tang et al., 2021) also adapt without source data, however, they optimize offline with full access to the whole target test data. In contrast, TTA does not need any specific modifications during the training phase and only requires the pre-trained source model and unlabeled target data during the testing phase, which is a more practical and feasible setting. To adapt a pre-trained model to an unlabeled target domain, a majority of TTA methods take inspiration from the semi-supervised learning field and employ various prevalent techniques tailored for unlabeled test data adaptation. Existing works of TTA mainly include batch norm calibration (Mirza et al., 2022; Schneider et al., 2020; Zhao et al., 2023; Gong et al., 2022; You et al., 2021a), consistency regularization (Boudiaf et al., 2022; Choi et al., 2022; Kojima et al., 2022; Döbler et al., 2023), and pseudo-labeling (Iwasawa & Matsuo, 2021; Jang et al., 2023; Li et al., 2023; Wang et al., 2023). Our work belongs to the pseudo-labeling based approach, which fine-tunes a pre-trained model using pseudo-labels based on classifier predictions. T3A (Iwasawa & Matsuo, 2021) designs the prototype-based classifier and predicts the labels of test data by comparing distances between the test data and the pseudo-prototypes. TAST (Jang et al., 2023) improves upon T3A by fine-tuning the pre-trained model through self-training with nearest neighbor information. In comparisons, we propose a graph based approach to incorporate both global representative features of prototypes and local nearest neighboring information.

### 2.2 SEMI-SUPERVISED LEARNING

Recent semi-supervised learning methods can be mainly categorized into consistency regularization based methods (Bachman et al., 2014; Sajjadi et al., 2016; Laine & Aila, 2016; French et al., 2017) and pseudo-labeling based methods (Lee et al., 2013; Rizve et al., 2021; Arazo et al., 2020; Wang et al., 2022a). Consistency regularization (CR) based methods assume that the output of the model should be invariant to random perturbations including data augmentation (French et al., 2017) and stochastic regularization (Laine & Aila, 2016; Sajjadi et al., 2016). Pseudo-labeling (PL) based methods use the model itself to produce artificial labels for unlabeled data. More recently, FixMatch (Sohn et al., 2020) produces artificial labels using both CR and PL to boost the performance of semi-supervised learning. However, it requires complicated augmentation designs of CR and handpicked thresholds of PL. In comparison, our proposed RST is simple and effective, which combines CR and PL in the context of TTA to handle noisy label learning.

Our proposed PGM is inspired by graph based semi-supervised learning which applies graph Laplacian regularization (Zhu et al., 2003; Yang et al., 2016) or contrastive regularization (Wan et al., 2021; Sun et al., 2019; 2020a) for representation learning with consistency regularization. However, these works primarily focus on modeling the relationship among labeled data and unlabeled data. In contrast, we build connections among prototypes and test data to obtain more reliable pseudo-labels. To the best of our knowledge, we are the first to apply graph based approach to TTA to facilitate effective message passing.

## 3 METHODOLOGY

### 3.1 PROBLEM DEFINITION

Test-time adaptation involves fine-tuning a source domain pre-trained model using the online unlabeled test data from the target domain. Given the model trained using standard empirical risk minimization (Chowdhary, 2020) on source data $\mathcal{D}_s$, a batch of unlabeled target data $\boldsymbol{x}_i \in \mathcal{D}_t$, where $i \in \{1, \ldots, N\}$ and $N$ denotes batch size, is sampled from the target domain $\mathcal{D}_t$ for domain adaptation. The model consists of a feature extractor $f_{\boldsymbol{\theta}}$ and a classifier $g_{\boldsymbol{\phi}}$. During testing, the model $h = f_{\boldsymbol{\theta}} \circ g_{\boldsymbol{\phi}}$ is initialized with pre-trained parameters. $K$ is the number of classes.

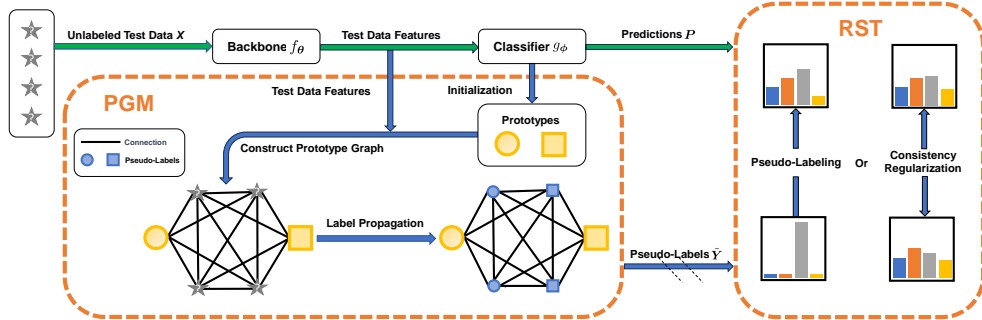

Figure 2: Overview of PROtotype GRAph Model based pseudo-label learning (PROGRAM). Green lines represent the forward and backward propagation, and blue lines represent only forward propagation. Different shapes of prototypes mean different classes. PGM: Prototype Graph Model. RST: Robust Self-Training.

## 3.2 OVERVIEW

In this section, we introduce our proposed method for test-time adaptation, termed PROtotype GRAph Model based pseudo-label learning (PROGRAM). PROGRAM consists of two main component including Prototype Graph Model (PGM) and Robust Self-Training (RST) as shown in Fig. 2. For each batch of test samples, PGM first initializes the prototypes according to the weights of classifier $g_\phi$, and constructs prototype graph among both prototypes and test samples via conditional probability distributions with respect to the prototypes. Label propagation is conducted in the constructed prototype graph and reliable pseudo-labels for the test samples are obtained. In RST, pseudo-labels or consistency regularization is utilized for model fine-tuning. When the hard labels of model predictions and pseudo-labels obtained by PGM are inconsistent, consistency loss is applied to exploit the noisy pseudo-labels. The final results are estimated using the model fine-tuned by RST. PROGRAM eliminates the interference of noisy pseudo-labels and adapts the source model to the target domain for robust predictions.

## 3.3 PROTOTYPE GRAPH MODEL (PGM)

**Prototype Graph Construction.** Similar to T3A (Iwasawa & Matsuo, 2021), we initialize prototypes using the model weights of classification layer as $c_k = \frac{w_k}{\|w_k\|_2}$, where $w_k$ is the $k$-th element of the weight matrix in the classifier $g_\phi$, representing the template of $k$-th class. For each test sample $x_i$, we compute a soft label $s_i = [s_{i1}, s_{i2}, ..., s_{iK}]^\top \in \mathbb{R}^K$ with regard to prototypes using the softmax function:

$$s_{ik} = \frac{\exp(c_k \cdot x_i')}{\sum_{k=1}^{K} \exp(c_k \cdot x_i')}, \tag{1}$$

where $x_i'$ denotes the feature representation of $x_i$ obtained from the feature extractor $f_\theta$. By calculating the soft labels of the entire batch of $N$ test samples, we obtain a sample label matrix $S = [s_1, s_2, ..., s_N]^\top \in \mathbb{R}^{N \times K}$ for the batch. In order to represent the labels for prototypes, we define a prototype label matrix $T \in \mathbb{R}^{K \times K}$, where one-hot encoding scheme is utilized that is $T_{ij} = 1$ if the $i$-th prototype corresponds to the class $j$, and $T_{ij} = 0$ otherwise. In general, $T$ is a diagonal matrix with ones on the diagonal when the order of prototypes is sequential. We concatenate $T$ and $S$ along the row axis to form the label matrix $Z \in \mathbb{R}^{(K+N) \times K}$.

Prototypes can be regarded as a good representation of the class centers. Instead of only considering the connections among test samples, we propose prototype graph to capture the relationships among test samples and prototypes. We construct a graph $\mathcal{G} = (\mathcal{V}, \mathcal{E})$, where the vertices $\mathcal{V}$ represent both the prototypes $v_i$ ($1 \le i \le K$) and the batch of unlabeled test samples $v_i$ ($K + 1 \le i \le K + N$), and the edges $\mathcal{E}$ modeling their relationship are represented by an adjacency matrix $W \in \mathbb{R}^{(K+N) \times (K+N)}$. We compute the similarity $w_{ij}$ between the vertex $v_i \in \mathcal{V}$ and vertex $v_j \in \mathcal{V}$ to determine their connectivity with respect to the prototypes $v_k$ ($1 \le k \le K$). In the prototype-based graph, we leverage a small number of prototypes to turn sample-to-sample affinity computations into

much simpler sample-to-prototype interactions. Similar ideas have also been exploited in (Zhu & Koniusz, 2023) for transductive few-shot learning. Specifically, the Markov random walks (Lovász, 1993; Szummer & Jaakkola, 2001) and Bayes' theorem are employed as follows:

$$w_{ij} = \text{similarity}(v_i, v_j) \propto p(v_i|v_j) = \sum_{k=1}^{K} p(v_i|v_k)p(v_k|v_j) = \sum_{k=1}^{K} z_{ik} \cdot \frac{z_{jk}}{\sum_{j'=1}^{K+N} z_{j'k}}. \quad (2)$$

We define $p(v_i|v_k) = z_{ik}$ where $1 \leq i \leq K + N$ and $1 \leq k \leq K$. $\boldsymbol{W}$ is a symmetric matrix and $w_{ij} = w_{ji}$. Given the diagonal matrix $\boldsymbol{D} \in \mathbb{R}^{K \times K}$ and $\boldsymbol{D}_{kk} = \sum_{i=1}^{K+N} z_{ik}$, the adjacency matrix $\boldsymbol{W}$ can be formulated as:

$$\boldsymbol{W} = \boldsymbol{Z}\boldsymbol{D}^{-1}\boldsymbol{Z}^{\top}. \quad (3)$$

Considering that prototypes are typically good representation of classes while neighboring samples may contain some noises, our proposed prototype graph is constructed by way of the prototypes, which is stable and effective.

**Prototype Graph based Label Propagation.** After prototype graph construction, we propose to propagate labels (Zhou et al., 2003) by optimizing the following problem:

$$\boldsymbol{Y}^* = \arg\min_{\boldsymbol{Y}} \frac{1}{2}(\sum_{i,j=1}^{K+N} w_{ij}\|\boldsymbol{y}_i - \boldsymbol{y}_j\|_2^2 + \mu \sum_{i=1}^{K+N} \|\boldsymbol{y}_i - \boldsymbol{z}_i\|_2^2). \quad (4)$$

$\boldsymbol{Y} = [\boldsymbol{y}_1, \boldsymbol{y}_2, ..., \boldsymbol{y}_K, ..., \boldsymbol{y}_{K+N}]^{\top} \in \mathbb{R}^{(K+N) \times K}$ is the optimization objective where $\boldsymbol{y}_i$ ($1 \leq i \leq K$) represents the optimized labels of the prototypes and $\boldsymbol{y}_i$ ($K + 1 \leq i \leq K + N$) represents those of test samples as pseudo-labels. $\boldsymbol{Y}^*$ denotes the optimal solution. $\boldsymbol{z}_i \in \mathbb{R}^K$ is the $i$-th row in the inital label matrix $\boldsymbol{Z}$.

The first term in Eq. 4 measures the *smoothness constraint*, which promotes smoothness by penalizing large differences between neighboring vertices in the graph. The second term is the *fitting constraint*, which measures the discrepancy between the optimized labels and the label assignment with regard to prototypes. The hyperparameter $\mu > 0$ controls the trade-off between the two terms.

The optimal solution can be obtained by the following equation (refer to Sec. A.1 for more details):

$$\boldsymbol{Y}^* = (1 - \lambda)\left(\mathbb{I} - \lambda\boldsymbol{W}\right)^{-1}\boldsymbol{Z} = (1 - \lambda)\left(\mathbb{I} - \lambda\boldsymbol{Z}\boldsymbol{D}^{-1}\boldsymbol{Z}^{\top}\right)^{-1}\boldsymbol{Z}, \quad (5)$$

where $\mathbb{I}$ is the identical matrix and $\lambda = \frac{1}{1+\mu}$ ($0 < \lambda < 1$) is the hyperparameter to balance the two constraints. Finally, we extract the pseudo-labels of the test samples $\tilde{\boldsymbol{Y}}$ by selecting the corresponding rows from the optimal solution $\boldsymbol{Y}^*$ and applying the softmax function: $\tilde{\boldsymbol{Y}} = \text{Softmax}(\boldsymbol{Y}^*_{K+1:K+N}) = [\tilde{\boldsymbol{y}}_{K+1}, \tilde{\boldsymbol{y}}_{K+2}, ..., \tilde{\boldsymbol{y}}_{K+N}]^{\top}$.

**Re-initializing Prototypes.** In our online test setting, for each batch of input test data, we re-initialize the prototype set leveraging the linear layer weights of the source model. This design ensures that prototypes always keep up to date and maintain their representation of global class characteristics, while minimizing the memory burden associated with prototype updates.

### 3.4 ROBUST SELF-TRAINING (RST)

To better utilize the pseudo-labels generated by PGM, we propose a method named Robust Self-Training (RST). Unlike the conventional approaches that directly use hard pseudo-labels as supervision, our method draws inspiration from FixMatch (Sohn et al., 2020) and combines both pseudo-labels and consistency regularization. Specifically, if the linear classifier and PGM produce the same predictions (*i.e.* identical results after applying argmax to the logits), we employ hard pseudo-labels. On the other hand, if they yield different predictions, we enforce consistency regularization by utilizing the Symmetric Cross Entropy (SCE) loss function (Wang et al., 2019):

$$L_{SCE} = -\sum_{k=1}^{K} \tilde{y}_{ik} \log p_{ik} - \sum_{k=1}^{K} p_{ik} \log \tilde{y}_{ik}, \quad (6)$$

---

**Algorithm 1** PROtotype GRAph Model based pseudo-label learning (PROGRAM)

---

**Input**: Pre-trained source model $h = f_{\boldsymbol{\theta}} \circ g_{\boldsymbol{\phi}}$, number of class $K$, batch size $N$, a test batch $\boldsymbol{x}$, cost balance coefficient $\lambda$, loss balance term $\beta$.

**Output**: Model prediction $\boldsymbol{y}_{pred}$.

   (1) PGM generates reliable pseudo-labels:

       Initialize prototypes: $\boldsymbol{c}_k = \frac{\boldsymbol{w}_k}{\|\boldsymbol{w}_k\|_2}$

       Prototype graph construction following Eq. 3:

$$\boldsymbol{x}_i' = f_{\boldsymbol{\theta}}(\boldsymbol{x}_i),\ s_{ik} = \frac{\boldsymbol{c}_k \cdot \boldsymbol{x}_i'}{\sum_{k=1}^{K} \boldsymbol{c}_k \cdot \boldsymbol{x}_i'},\ \boldsymbol{T} = \mathbb{I}_k,\ \boldsymbol{Z} = \mathrm{Concat}(\boldsymbol{T}, \boldsymbol{S}),\ \boldsymbol{D}_{kk} = \sum_{i=1}^{K+N} z_{ik}$$
$$\boldsymbol{W} = \boldsymbol{Z}\boldsymbol{D}^{-1}\boldsymbol{Z}^{\top}.$$

       Prototype graph label propagation following Eq. 5:

$$\boldsymbol{Y}^* = (1-\lambda)\left(\mathbb{I} - \lambda \boldsymbol{Z}\boldsymbol{D}^{-1}\boldsymbol{Z}^{\top}\right)^{-1} \boldsymbol{Z},\ \tilde{\boldsymbol{Y}} = \mathrm{Softmax}(\boldsymbol{Y}^*_{K+1:K+N})$$

   (2) RST fine-tunes the pre-trained model with Eq. 7 to obtain fine-tuned model $h_{ft}$.

$$L_{RST} = \sum_{i=1}^{N} \begin{cases} -\sum_{k=1}^{K} \arg\max \boldsymbol{y}_i \log p_{ik} & \text{if } \arg\max \boldsymbol{p}_i = \arg\max \boldsymbol{y}_i \\ \beta(L_{SCE}) & \text{else} \end{cases}$$

   (3) Get final predictions:

       $\boldsymbol{y}_{pred} = h_{ft}(\boldsymbol{x})$.

**Return**: $\boldsymbol{y}_{pred}$

---

where $\boldsymbol{p}_i = [p_{i1}, p_{i2}, ..., p_{iK}]^{\top} \in \mathbb{R}^K$ represents the inferred output distribution of the $i$-th test sample, and $\tilde{\boldsymbol{y}}_i = [\tilde{y}_{i1}, \tilde{y}_{i2}, ..., \tilde{y}_{iK}]^{\top} \in \mathbb{R}^K$ denotes the pseudo-label generated by PGM. This loss function serves as a consistency loss function that exhibits robustness to noisy labels.

Consequently, we formulate the final training objective function as follows:

$$L_{RST} = \sum_{i=1}^{N} \begin{cases} -\sum_{k=1}^{K} \arg\max \boldsymbol{y}_i \log p_{ik} & \text{if } \arg\max \boldsymbol{p}_i = \arg\max \boldsymbol{y}_i \\ \beta(L_{SCE}) & \text{else} \end{cases}, \tag{7}$$

where $\beta$ is the loss balance term and we empirically set it as $\beta = 0.4$. This loss function enables all pseudo-labels to fine-tune the source model and sufficiently mitigates the negative impact of noisy pseudo-labels. Theoretical analysis is presented in Sec. A.3. Furthermore, the alternating use of the two loss functions helps prevent model collapse caused by consistency loss (Jing et al., 2021).

## 3.5 ALGORITHM SUMMARY

We summarize the pipeline of PROGRAM in Algorithm 1. During the testing phase, the adaptation is performed in an online manner. For each batch of test data, PGM first initializes prototypes, and constructs the prototype graph for label propagation. After that, RST is applied to fine-tune the pre-trained model. Finally, the fine-tuned model is used to produce prediction results.

## 4 EXPERIMENTS

### 4.1 EXPERIMENTAL SETUP

Due to space limit, more details about the experimental setup including datasets (Sec. F.1), models (Sec. F.2), baselines (Sec. F.3) and implementation (Sec. F.4) are provided in Appendix.

**Datasets.** We conduct experiments on four domain generalization benchmarks (*i.e.*, VLCS (Torralba & Efros, 2011), PACS (Li et al., 2017), OfficeHome (Venkateswara et al., 2017), and TerraIncognita (Beery et al., 2018)) and three image corruption benchmarks (*i.e.*, CIFAR-10/100C (Hendrycks & Dietterich, 2019) and Imagenet-C (Hendrycks & Dietterich, 2019)). For domain generalization datasets, we choose one domain as the target domain and the others as the source domains. For image corruption datasets, we select the original CIFAR-10/100 (Krizhevsky, 2009) and ImageNet (Krizhevsky et al., 2012) as the source domains. And the corrupted test data as the target domain. We follow the dataset splits as TAST (Jang et al., 2023) for a fair comparison.

**Models.** In the main experiments, we compare different methods on ResNet-18/50 (He et al., 2016) backbones, which are widely used in domain adaptation and generalization community. Also, we

Table 1: Comparisons with the state-of-the-art methods with *average accuracy* (%) on four domain generalization benchmarks. Avg. is the average performance of all the datasets. + indicates the ERM baseline combined with the respective TTA method. ↑ means higher is better, and * denotes the results from (Jang et al., 2023).

| Method | Backbone | VLCS ↑ | PACS ↑ | OfficeHome ↑ | TerraIncognita ↑ | Avg. ↑ |
|---|---|---|---|---|---|---|
| ERM* (Chowdhary, 2020) | ResNet-18 | 74.88±0.46 | 79.29±0.77 | 62.10±0.31 | 40.62±1.19 | 64.22 |
| +Tent* (Wang et al., 2021a) | ResNet-18 | 72.88±0.82 | 83.89±0.54 | 60.86±0.39 | 33.70±1.09 | 62.83 |
| +TentAdapter* (Wang et al., 2021a) | ResNet-18 | 67.02±1.16 | 80.75±1.01 | 62.64±0.38 | 39.91±0.76 | 62.58 |
| +TentClf* (Wang et al., 2021a) | ResNet-18 | 72.96±1.48 | 78.57±1.78 | 59.33±0.62 | 38.30±3.44 | 62.29 |
| +SHOT* (Liang et al., 2020) | ResNet-18 | 65.24±2.29 | 82.36±0.63 | 62.58±0.39 | 33.57±1.04 | 60.94 |
| +SHOTIM* (Liang et al., 2020) | ResNet-18 | 64.86±2.22 | 82.33±0.61 | 62.57±0.39 | 33.35±1.23 | 60.78 |
| +PL* (Lee et al., 2013) | ResNet-18 | 62.97±2.72 | 70.98±1.78 | 58.20±3.21 | 37.44±7.20 | 57.40 |
| +PLClf* (Lee et al., 2013) | ResNet-18 | 74.89±0.61 | 78.11±2.30 | 61.92±0.41 | 41.78±1.94 | 64.18 |
| +T3A* (Iwasawa & Matsuo, 2021) | ResNet-18 | 77.26±1.49 | 80.83±0.67 | 63.21±0.50 | 40.20±0.60 | 65.38 |
| +TAST* (Jang et al., 2023) | ResNet-18 | 77.27±0.67 | 81.94±0.44 | 63.70±0.52 | 42.64±0.72 | 66.39 |
| +TAST-BN* (Jang et al., 2023) | ResNet-18 | 75.21±2.36 | 87.07±0.53 | 62.79±0.41 | 39.43±2.24 | 66.13 |
| +TSD (Wang et al., 2023) | ResNet-18 | 73.57±1.08 | 87.06±0.68 | 64.51±1.22 | 42.65±1.47 | 66.95 |
| +PROGRAM | ResNet-18 | **77.75±1.37** | **88.03±0.74** | **64.59±0.85** | **43.16±1.12** | **68.38** |
| ERM* (Chowdhary, 2020) | ResNet-50 | 76.71±0.50 | 83.21±1.14 | 67.13±0.99 | 45.93±1.34 | 68.25 |
| +Tent* (Wang et al., 2021a) | ResNet-50 | 72.96±1.27 | 85.16±0.62 | 66.29±0.77 | 37.08±2.04 | 65.37 |
| +TentAdapter* (Wang et al., 2021a) | ResNet-50 | 69.65±1.17 | 83.69±1.16 | 67.91±0.89 | 43.89±1.25 | 66.29 |
| +TentClf* (Wang et al., 2021a) | ResNet-50 | 75.80±0.68 | 82.66±1.59 | 66.79±0.98 | 43.64±2.59 | 67.22 |
| +SHOT* (Liang et al., 2020) | ResNet-50 | 67.07±0.90 | 84.07±1.23 | 67.65±0.72 | 35.20±0.82 | 63.50 |
| +SHOTIM* (Liang et al., 2020) | ResNet-50 | 66.93±0.84 | 84.14±1.25 | 67.65±0.77 | 34.37±1.07 | 63.27 |
| +PL* (Lee et al., 2013) | ResNet-50 | 69.41±3.12 | 81.72±4.61 | 62.85±3.05 | 38.09±2.35 | 63.02 |
| +PLClf* (Lee et al., 2013) | ResNet-50 | 75.65±0.88 | 83.33±1.59 | 67.01±1.00 | 46.66±2.12 | 68.16 |
| +T3A* (Iwasawa & Matsuo, 2021) | ResNet-50 | 77.29±0.39 | 83.92±1.13 | 68.26±0.84 | 45.61±1.10 | 68.77 |
| +TAST* (Jang et al., 2023) | ResNet-50 | 77.66±0.48 | 84.11±1.22 | 68.63±0.70 | 47.43±2.09 | 69.46 |
| +TAST-BN* (Jang et al., 2023) | ResNet-50 | 73.52±1.37 | 89.16±0.47 | 68.88±0.50 | 41.47±2.88 | 68.26 |
| +TSD (Wang et al., 2023) | ResNet-50 | 74.46±0.93 | 89.27±0.59 | 68.43±0.74 | 47.41±1.60 | 69.89 |
| +PROGRAM | ResNet-50 | **78.17±0.92** | **90.78±0.40** | **69.05±0.66** | **47.84±1.23** | **71.46** |

Table 2: Comparisons with the state-of-the-art methods with *average error rate* (%) on image corruption benchmarks. Testing is conducted on the highest level of image corruption. All methods use ResNet-50 backbone. ↓ means lower is better. * denotes the results from (Jang et al., 2023).

| Method | CIFAR-10C ↓ | CIFAR-100C ↓ | ImageNet-C ↓ | Avg. ↓ |
|---|---|---|---|---|
| No Adaptation* (Chowdhary, 2020) | 29.14 | 60.35 | 81.99 | 57.16 |
| +SHOT* (Liang et al., 2020) | 15.32 | 41.54 | 58.27 | 38.38 |
| +Tent* (Wang et al., 2021a) | 13.95 | 39.04 | 58.06 | 37.02 |
| +PL* (Lee et al., 2013) | 22.34 | 40.06 | 62.95 | 41.78 |
| +T3A* (Iwasawa & Matsuo, 2021) | 26.68 | 58.28 | 75.81 | 53.59 |
| +TAST* (Jang et al., 2023) | 26.61 | 60.74 | - | - |
| +TAST-BN* (Jang et al., 2023) | 13.08 | 37.82 | 67.05 | 39.32 |
| +TIPI (Nguyen et al., 2023) | 13.52 | 38.33 | 55.94 | 35.93 |
| +TSD (Wang et al., 2023) | 13.05 | 37.67 | 53.18 | 34.63 |
| +PROGRAM | **11.91** | **36.42** | **51.43** | **33.25** |

test our method on different backbones, including ViT-B/16 (Dosovitskiy et al., 2020), ResNeXt-50 (Xie et al., 2017), EfficientNet-B4 (Niu et al., 2022), and Mixer-L/16 (Tolstikhin et al., 2021).

**Baselines.** We compare PROGRAM with the following baselines: Empirical Risk Minimization (ERM) (Chowdhary, 2020), PL and PLClf (Lee et al., 2013), Tent, TentAdapter and TentClf (Wang et al., 2021a), SHOTIM and SHOT (Liang et al., 2020), T3A (Iwasawa & Matsuo, 2021), TAST and TAST-BN (Jang et al., 2023), TSD (Wang et al., 2023), and TIPI (Nguyen et al., 2023). For fair comparisons, all methods are based on online batch-level test data adaptation setting.

**Implementation.** During the test phase, we use the Adam optimizer (Loshchilov & Hutter, 2017) to fine-tune the whole network parameters. We set the cost balance coefficient $\lambda = 0.5$ and loss balance term $\beta = 0.4$. We report the mean value and variance of results with four different random seeds $\{0, 1, 2, 3\}$ and data splits on domain generalization benchmarks.

## 4.2 EXPERIMENTAL RESULTS

**Domain Generalization Benchmarks.** Table 1 shows the results with ResNet-18/50 backbones on four popular domain generalization datasets. Average accuracy (top-1) is reported. We follow

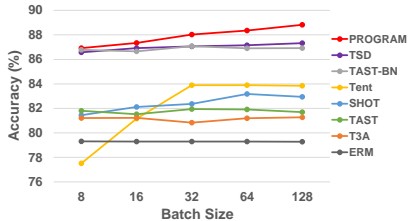

Figure 3: Effect of batch sizes on PACS dataset with ResNet-18 backbone.

Table 3: Effect of Prototype Graph Model (PGM) and Robust Self-Training (RST) on the domain generalization datasets. *Average accuracy* (%) is reported.

| Method | VLCS ↑ | PACS ↑ | OfficeHome ↑ | TerraIncognita ↑ | Avg. ↑ |
|---|---|---|---|---|---|
| ResNet-18 | 74.88±0.46 | 79.29±0.77 | 62.10±0.31 | 40.62±1.19 | 64.22 |
| +PGM | 77.12±0.87 | 87.51±1.00 | 63.66±0.78 | 41.94±0.95 | 67.56 |
| +PGM + RST | 77.75±1.37 | 88.03±0.74 | 64.59±0.85 | 43.16±1.12 | 68.38 |
| ResNet-50 | 76.71±0.50 | 83.21±1.14 | 67.13±0.99 | 45.93±1.34 | 68.25 |
| +PGM | 77.72±0.83 | 90.45±0.37 | 68.23±0.78 | 46.92±1.04 | 70.83 |
| +PGM + RST | 78.17±0.92 | 90.78±0.40 | 69.05±0.66 | 47.84±1.23 | 71.46 |

Table 4: Applying Prototype Graph Model (PGM) to pseudo-labeling based approaches. *Average accuracy* (%) is reported on domain generalization benchmarks, and *average error rate* (%) is reported on image corruption benchmarks. ↑ means higher is better, while ↓ means lower is better.

| Method | Domain Generalization Benchmark | | | | | Image Corruption Benchmark | |
|---|---|---|---|---|---|---|---|
| | VLCS ↑ | PACS ↑ | OfficeHome ↑ | TerraIncognita ↑ | Avg. ↑ | CIFAR-10C ↓ | CIFAR-100C ↓ |
| T3A (Iwasawa & Matsuo, 2021) | 77.29±0.39 | 83.92±1.13 | 68.26±0.84 | 45.61±1.10 | 68.77 | 26.68 | 58.28 |
| T3A + PGM | 78.51±0.57 | 85.43±1.28 | 68.92±1.25 | 46.26±1.38 | 69.78 (+1.01) | 25.15 (-1.53) | 57.06 (-1.22) |
| TAST (Jang et al., 2023) | 77.66±0.48 | 84.11±1.22 | 68.63±0.70 | 47.43±2.09 | 69.46 | 26.61 | 60.74 |
| TAST + PGM | 78.38±0.78 | 85.41±1.49 | 69.03±0.92 | 48.42±1.79 | 70.31 (+0.85) | 25.34 (-1.27) | 59.65 (-1.09) |
| TAST-BN (Jang et al., 2023) | 73.52±1.37 | 89.16±0.47 | 68.88±0.50 | 41.47±2.88 | 68.26 | 13.08 | 37.82 |
| TAST-BN + PGM | 74.56±1.74 | 90.49±0.72 | 69.47±1.02 | 42.09±2.53 | 69.15 (+0.89) | 12.44 (-0.64) | 36.79 (-1.03) |

the common setting (Jang et al., 2023) to set the batch size as 32. In Table 1, we observe that our proposed PROGRAM consistently improves upon the ERM baseline and achieves the state-of-the-art performance on all the datasets. Our method improves ERM by 4.16% on average with ResNet-18 backbone and 3.21% with ResNet-50 backbone. In comparison, other TTA methods do not produce consistent improvement on all datasets. Full results are provided in Appendix H.

**Image Corruption Benchmarks.** Image corruption benchmarks are designed to evaluate the robustness and generalization ability of a classifier on unseen corrupted samples which is pre-trained using clean data. In Table 2, we compare PROGRAM with other TTA methods on CIFAR-10C/100C and ImageNet-C benchmarks. Average error rate (top-1) is reported. ResNet-50 is used as the backbone of all the methods. We follow TAST (Jang et al., 2023) to set the test batch size as 128 on the CIFAR-10C/100C datasets and 64 on the ImagNet-C dataset. Experimental results show that our proposed PROGRAM reduces the average error by 23.91% on average, significantly outperforming all the other TTA methods. Please refer to Appendix H for the details of full results.

### 4.3 ANALYSIS

**Different Batch Sizes.** In Fig. 3, we report the average accuracy of various methods under different batch sizes on the PACS dataset with the ResNet-18 backbone. We observe that our approach outperforms other methods under different batch sizes. With the increase of batch size, the superiority of our proposed PROGRAM becomes more remarkable. More analysis can be found in Sec. D.1.

**Effect of Model Components.** In Table 3, we conduct ablation studies to validate the effectiveness of the proposed Prototype Graph Model (PGM) and Robust Self-Training (RST) on the domain generalization datasets. Both PGM and RST consistently improve the performance of ResNet-18/50 backbones on all the datasets. Furthermore, we show that our PGM is "plug-and-play" for pseudo-label generation in Table 4. PGM is applied to typical pseudo-labeling based methods (*i.e.*, T3A (Iwasawa & Matsuo, 2021) and TAST (Jang et al., 2023)) and replace their pseudo-label generation stage (refer to Appendix G for more details). We report the results with ResNet-50 backbone on domain generalization datasets and image corruption datasets. PGM improves T3A and TAST by a large margin, which demonstrates that PGM generates more reliable pseudo-labels compared with other pseudo-labeling approaches. More experiments can be found in Sec. D.2, D.3 and D.4.

**Different Backbones.** In Table 5, we validated our method on various backbone architectures with the test batch size of 128 following TSD (Wang et al., 2023). Experimental results show that our proposed PROGRAM consistently improves upon different model architectures, including ViT (Dosovitskiy et al., 2020), ResNeXt (Xie et al., 2017), EfficientNet (Tan & Le, 2019), and MLP-

Table 5: Results (*average accuracy*) with different backbones. All baseline models are trained in a standard ERM manner. ↑ means higher is better, and * denotes the results from (Wang et al., 2023).

| Backbone | PACS ↑ | OfficeHome ↑ | VLCS ↑ |
|---|---|---|---|
| ViT-B/16* | 87.13 | 79.06 | 78.70 |
| +TSD* (Wang et al., 2023) | 90.20 | 81.80 | 79.90 |
| +PROGRAM | **91.96** | **82.63** | **83.08** |
| ResNeXt-50* | 86.67 | 72.66 | 78.50 |
| +TSD* (Wang et al., 2023) | 91.33 | 74.18 | 79.38 |
| +PROGRAM | **92.84** | **74.89** | **82.24** |
| EfficientNet-B4* | 85.11 | 74.65 | 77.14 |
| +TSD* (Wang et al., 2023) | 85.41 | 72.24 | 79.42 |
| +PROGRAM | **86.78** | **74.71** | **82.35** |
| Mixer-L/16* | 84.59 | 71.36 | 76.53 |
| +TSD* (Wang et al., 2023) | 88.47 | 74.82 | 79.75 |
| +PROGRAM | **90.29** | **75.46** | **82.88** |

Table 6: Runtime analysis with ResNet-18 backbone on a single Titan XP GPU on the PACS dataset.

| | Method | Runtime (s) |
|---|---|---|
| Partial | TentClf (Wang et al., 2021a) | 0.40 |
| | TentAdapter (Wang et al., 2021a) | 0.43 |
| | Tent (Wang et al., 2021a) | 15.17 |
| | PLClf (Lee et al., 2013) | 0.43 |
| | PL (Lee et al., 2013) | 20.75 |
| | T3A (Iwasawa & Matsuo, 2021) | 0.58 |
| | TAST (Jang et al., 2023) | 6.92 |
| | TAST-BN (Jang et al., 2023) | 73.93 |
| Whole | SHOT (Liang et al., 2020) | 20.97 |
| | SHOTIM (Liang et al., 2020) | 20.73 |
| | TSD (Wang et al., 2023) | 19.73 |
| | PROGRAM | 17.92 |

mixer (Tolstikhin et al., 2021). Compared with TSD (Wang et al., 2023), PROGRAM achieves better performance across all the model architectures under the same setting.

**Runtime Analysis.** In Table 6, we report the speed of TTA methods with ResNet-18 backbone on the PACS dataset, which is the average runtime of fine-tuning the model using a batch of test samples with batch size 32 on a Titan XP GPU. Compared with these methods that update the "whole" feature extractors (*e.g.*, SHOT (Liang et al., 2020) and TSD (Wang et al., 2023)), the efficiency of PROGRAM is comparable. Please refer to Sec. C for more details. We also present a partial-update variant of PROGRAM in Sec. D.6 to explore the trade-off between effectiveness and efficiency.

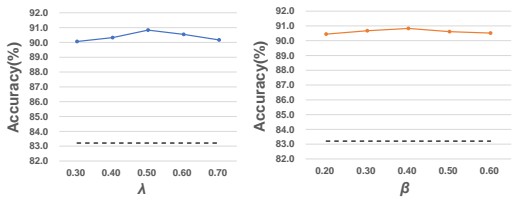 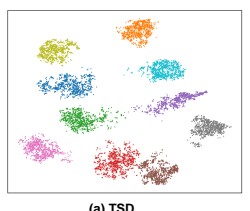 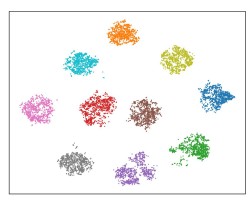

Figure 4: Sensitivity analysis regarding $\lambda$ and $\beta$ on PACS dataset. Accuracy of ERM baseline is shown with the dotted black lines. Our PROGRAM is robust to different hyperparaters.

Figure 5: t-SNE visualization of learned feature embeddings extracted by (a) TSD and (b) PROGRAM on CIFAR-10C dataset. Different colors denote different classes.

**Sensitivity to hyper-parameters.** We investigate the sensitivity of PROGRAM to the cost balance coefficient $\lambda$ (*cf.* Eq. 5) and the loss balance term $\beta$ (*cf.* Eq. 7). As shown in Fig. 4, our method is insensitive to the hyper-parameters and consistently improves upon the ERM baseline with different hyper-parameters. In our implementation, we choose $\lambda = 0.5$ and $\beta = 0.4$ for best performance.

**Qualitative Analysis.** In Fig. 5, we visualize the t-SNE (Maaten & Hinton, 2008) of feature embeddings extracted by the fine-tuned models on CIFAR-10C dataset. The TSD (Wang et al., 2023) learned features of different categories are not well separated, and some test samples are hard to distinguish due to the large domain shift. In comparison, our method generates compact and discriminative feature embeddings with tight clusters, demonstrating the superiority of PROGRAM.

## 5 CONCLUSIONS

In this work, we propose a novel pseudo-labeling based TTA method, termed PROtotype GRAph Model based pseudo-label learning (PROGRAM). In the pseudo-label generation stage, we propose Prototype Graph Model (PGM) that combines the superiority of prototype based and nearest neighbor based methods to produce reliable pseudo-labels. In the self-training stage, Robust Self-Training (RST) is applied for test-time adaptation to resist noisy pseudo-labels. Extensive experiments demonstrate that our proposed PROGRAM consistently outperforms the existing methods on various domain generalization and image corruption benchmarks. Besides, we show that PROGRAM is "plug-and-play" and can be easily integrated into different backbone networks and various TTA methods.

**Acknowledgement.** This paper is partially supported by the National Key R&D Program of China No.2022ZD0161000 and the General Research Fund of Hong Kong No.17200622.

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

## A  THEORETICAL ANALYSIS

### A.1  DERIVATION OF EQUATIONS FOR PROTOTYPE GRAPH BASED LABEL PROPAGATION

We rewrite Eq. 4 as follows:

$$\mathcal{L}(\boldsymbol{Y}) = \frac{1}{2}\Big(\sum_{i,j=1}^{K+N} w_{ij}\|\boldsymbol{y}_i - \boldsymbol{y}_j\|_2^2 + \mu \sum_{i=1}^{K+N} \|\boldsymbol{y}_i - \boldsymbol{z}_i\|_2^2\Big). \tag{A1}$$

This defines the cost function $\mathcal{L}$ associated with $\boldsymbol{Y}$.

$$\boldsymbol{Y}^* = \arg\min_{\boldsymbol{Y}} \mathcal{L}(\boldsymbol{Y}). \tag{A2}$$

Differentiating $\mathcal{L}(\boldsymbol{Y})$ with respect to $\boldsymbol{Y}$, we have

$$\frac{\partial \mathcal{L}}{\partial \boldsymbol{Y}}\Big|_{\boldsymbol{Y}=\boldsymbol{Y}^*} = \boldsymbol{Y}^* - \boldsymbol{W}\boldsymbol{Y}^* + \mu(\boldsymbol{Y}^* - \boldsymbol{Z}) = 0. \tag{A3}$$

We expand the parentheses at the end to obtain the following expression:

$$\frac{\partial \mathcal{L}}{\partial \boldsymbol{Y}}\Big|_{\boldsymbol{Y}=\boldsymbol{Y}^*} = (1 + \mu)\boldsymbol{Y}^* - \boldsymbol{W}\boldsymbol{Y}^* - \mu\boldsymbol{Z} = 0. \tag{A4}$$

Dividing both sides of the equation by $(1 + \mu)$, we get:

$$\boldsymbol{Y}^* - \frac{1}{1+\mu}\boldsymbol{W}\boldsymbol{Y}^* - \frac{\mu}{1+\mu}\boldsymbol{Z} = 0. \tag{A5}$$

Introducing a new variable $\lambda = \frac{1}{1+\mu}$, we get

$$(\mathbb{I} - \lambda\boldsymbol{W})\boldsymbol{Y}^* = (1 - \lambda)\boldsymbol{Z}. \tag{A6}$$

Since $(\mathbb{I} - \lambda\boldsymbol{W})$ is generally invertible (see details in Sec. A.2), we have

$$\boldsymbol{Y}^* = (1 - \lambda)(\mathbb{I} - \lambda\boldsymbol{W})^{-1}\boldsymbol{Z}. \tag{A7}$$

### A.2  MATRIX INVERTIBILITY PROOF

In this section, we provide a detailed proof of the invertibility of $(\mathbb{I} - \lambda\boldsymbol{W})$.

$(\mathbb{I} - \lambda\boldsymbol{W})$ can be rewritten as:

$$\mathbb{I} - \lambda\boldsymbol{W} = (1 - \lambda)\mathbb{I} + \lambda(\mathbb{I} - \boldsymbol{W}). \tag{A8}$$

As $\lambda = \frac{1}{1+\mu}$ is a number greater than 0 and less than 1, matrix $(1 - \lambda)\mathbb{I}$ is a positive definite matrix. And the matrix $\mathbb{I} - \boldsymbol{W}$ is a Laplacian matrix (we will prove it later), which is a positive semidefinite matrix. Therefore, the matrix $\mathbb{I} - \lambda\boldsymbol{W} = (1 - \lambda)\mathbb{I} + \lambda(\mathbb{I} - \boldsymbol{W})$ is a positive definite matrix. Furthermore, due to the symmetry of the matrix, $(\mathbb{I} - \lambda\boldsymbol{W})$ is invertible.

**Proof of Laplacian matrix.** In the following section, we will prove $\mathbb{I} - \boldsymbol{W}$ is a Laplacian matrix, and thus is positive semidefinite.

Starting from Eq. 3, we express $(\mathbb{I} - \lambda\boldsymbol{W})$ as:

$$(\mathbb{I} - \lambda\boldsymbol{W}) = \mathbb{I} - \lambda\boldsymbol{Z}\boldsymbol{D}^{-1}\boldsymbol{Z}^T. \tag{A9}$$

Given the matrices:

$$\boldsymbol{Z} = \begin{bmatrix} \boldsymbol{T}_{K\times K} \\ \boldsymbol{S}_{N\times K} \end{bmatrix}, \tag{A10}$$

$(\boldsymbol{T}_{K\times K} = \mathbb{I}_{K\times K})$ and

$$\boldsymbol{Z}\boldsymbol{D}^{-1}\boldsymbol{Z}^T = \begin{bmatrix} \boldsymbol{D}_{K \times K}^{-1} & \boldsymbol{D}_{K \times K}^{-1}\boldsymbol{S}_{K \times N}^T \\ \boldsymbol{S}_{N \times K}\boldsymbol{D}_{K \times K}^{-1} & \boldsymbol{S}_{N \times K}\boldsymbol{D}_{K \times K}^{-1}\boldsymbol{S}_{K \times N}^T \end{bmatrix}. \tag{A11}$$

Let sum_row($\boldsymbol{X}, i$) denote the sum of the $i_{\text{th}}$ row in the matrix $\boldsymbol{X}$, and sum_col($\boldsymbol{X}, i$) represent the sum of the $i_{\text{th}}$ column in the matrix $\boldsymbol{X}$. According to the definition in the main text, $\boldsymbol{D}$ be a diagonal matrix, and the diagonal element $\boldsymbol{D}_{ii}$ signifies the sum of the matrix $\boldsymbol{Z}$ $i$-th column, expressed as:

$$\boldsymbol{D}_{ii} = \text{sum\_col}(\boldsymbol{T}, i) + \text{sum\_col}(\boldsymbol{S}, i) = 1 + \text{sum\_col}(\boldsymbol{S}, i). \tag{A12}$$

According to the definition in the main text, sum_row($\boldsymbol{S}, i$) = 1. For $0 \leq i \leq K - 1$, we compute the sum of the $i$-th row of $\boldsymbol{W}$ as follows:

$$\begin{aligned}
\text{sum\_row}(\boldsymbol{W}, i) &= \text{sum\_row}(\boldsymbol{D}^{-1}, i) + \text{sum\_row}(\boldsymbol{D}^{-1}\boldsymbol{S}^T, i) \\
&= \frac{1}{\boldsymbol{D}_{ii}} + \frac{\text{sum\_row}(\boldsymbol{S}^T, i)}{\boldsymbol{D}_{ii}} \\
&= \frac{1}{\boldsymbol{D}_{ii}} + \frac{\text{sum\_col}(\boldsymbol{S}, i)}{\boldsymbol{D}_{ii}} \\
&= 1
\end{aligned} \tag{A13}$$

For $i \geq K$, we have:

$$\begin{aligned}
\text{sum\_row}(\boldsymbol{W}, i) &= \text{sum\_row}(\boldsymbol{S}\boldsymbol{D}^{-1}, i) + \text{sum\_row}(\boldsymbol{S}\boldsymbol{D}^{-1}\boldsymbol{S}^T, i) \\
&= \sum_{m=0}^{K-1} \frac{(\boldsymbol{S}_{im})}{\boldsymbol{D}_{mm}} + \sum_{n=0}^{k-1}\sum_{m=0}^{K-1} \frac{(\boldsymbol{S}_{im} \cdot \boldsymbol{S}_{nm})}{\boldsymbol{D}_{mm}} \\
&= \sum_{m=0}^{K-1} \frac{(\boldsymbol{S}_{im})}{\boldsymbol{D}_{mm}} + \sum_{m=0}^{K-1} \frac{(\boldsymbol{S}_{im} \cdot \sum_{n=0}^{k-1} \boldsymbol{S}_{nm})}{\boldsymbol{D}_{mm}} \\
&= \sum_{m=0}^{K-1} \frac{(\boldsymbol{S}_{im})}{\boldsymbol{D}_{mm}} + \sum_{m=0}^{K-1} \frac{(\boldsymbol{S}_{im} \cdot \text{sum\_col}(\boldsymbol{S}, m))}{\boldsymbol{D}_{mm}} \\
&= \sum_{m=0}^{K-1} \frac{(\boldsymbol{S}_{im} \cdot (1 + \text{sum\_col}(\boldsymbol{S}, m)))}{\boldsymbol{D}_{mm}} \\
&= \sum_{m=0}^{K-1} \boldsymbol{S}_{im} \\
&= 1
\end{aligned} \tag{A14}$$

Thus, the sum of rows in the matrix $\boldsymbol{W}$ is always 1. Therefore, the diagonal elements of $\mathbb{I}$ and the sum of elements in each row of $\boldsymbol{W}$ are equal. According to Merris (1994), the definition of the Laplacian matrix is as follows: "The function $\boldsymbol{C}$ is most conveniently described as an $n \times n$, symmetric, nonnegative matrix $\boldsymbol{C} = (c_{ij})$ with the property that $c_{ij} > 0$ if and only if $(v_i, v_j) \in E$. With $r_i$ denoting the $i$-th row sum of $C$, define $L(G) = \text{diag}(r_1, r_2, \ldots, r_n) - \boldsymbol{C}$."

The matrix $\mathbb{I} - \boldsymbol{W}$ satisfies the above requirements; thus, it is a Laplacian matrix. And Laplacian matrices are always positive semidefinite, which is pointed out by Merris (1994).

## A.3 THEORETICAL ANALYSIS ABOUT RST

In this section, we present a theoretical analysis to elucidate the effectiveness of our RST. For a given test sample $x$, we denote the probability distribution of the output class $k$ predicted by the model as $p_k = p(k|x)$, and the probability distribution predicted by PGM as $q_k = q(k|x)$. The total number of classes is $K$.

The gradient of the $L_{SCE}$ loss with respect to the logits $z_j$ can be derived as follows:

$$\frac{\partial \mathcal{L}_{SCE}}{\partial z_j} = -\sum_{k=1}^{K} \frac{q_k}{p_k} \frac{\partial p_k}{\partial z_j} - \sum_{k=1}^{K} \frac{\partial p_k}{\partial z_j} \log q_k, \tag{A15}$$

where

$$\frac{\partial p_k}{\partial z_j} = \frac{\partial \left( \frac{e^{z_k}}{\sum_{j=1}^{K} e^{z_j}} \right)}{\partial z_j} = \frac{\frac{\partial e^{z_k}}{\partial z_j} \left( \sum_{j=1}^{K} e^{z_j} \right) - e^{z_k} \frac{\partial (\sum_{j=1}^{K} e^{z_j})}{\partial z_j}}{\left( \sum_{j=1}^{K} e^{z_j} \right)^2}. \tag{A16}$$

In the case of $k = j$, we have:

$$\frac{\partial p_k}{\partial z_j} = \frac{\partial p_k}{\partial z_k} = \frac{e^{z_k}(\sum_{k=1}^{K} e^{z_k}) - (e^{z_k})^2}{\left( \sum_{k=1}^{K} e^{z_k} \right)^2} = \frac{e^{z_k}}{\sum_{k=1}^{K} e^{z_k}} - \left( \frac{e^{z_k}}{\sum_{k=1}^{K} e^{z_k}} \right)^2 = p_k - p_k^2 = p_k(1 - p_k). \tag{A17}$$

In the case of $k \neq j$, we have:

$$\frac{\partial p_k}{\partial z_j} = \frac{0 \cdot \left( \sum_{j=1}^{K} e^{z_j} \right) - e^{z_k} e^{z_j}}{\left( \sum_{j=1}^{K} e^{z_j} \right) \left( \sum_{j=1}^{K} e^{z_j} \right)} = -\frac{e^{z_k}}{\sum_{j=1}^{K} e^{z_j}} \frac{e^{z_j}}{\sum_{j=1}^{K} e^{z_j}} = -p_k p_j. \tag{A18}$$

Simplifying the expression, we obtain:

$$\begin{aligned} \frac{\partial \mathcal{L}_{SCE}}{\partial z_j} &= -\sum_{k=1}^{K} \frac{q_k}{p_k} \frac{\partial p_k}{\partial z_j} - \sum_{k=1}^{K} \frac{\partial p_k}{\partial z_j} \log q_k \\ &= -\sum_{k \neq j}^{K} \frac{q_k}{p_k}(-p_j p_k) - \frac{q_j}{p_j}(p_j(1 - p_j)) - \sum_{k \neq j}^{K}(-p_j p_k) \log q_k - p_j(1 - p_j) \log q_j \\ &= (p_j - q_j) + p_j \left( \sum_{k=1}^{K} p_k \log q_k - \log q_j \right). \end{aligned} \tag{A19}$$

First, we analyze the gradient of the first term in the SCE loss function, referring to it as $L_{CR} = p_j - q_j$. The gradient of consistency regularization $L_{CR}$ is:

$$\frac{\partial L_{CR}}{\partial z_j} = \begin{cases} p_j - q_j \leq 0, & p_j \leq q_j \\ p_j - q_j > 0, & p_j > q_j \end{cases} \tag{A20}$$

When the $p_j$ for category $j$ is greater than $q_j$, the gradient is positive, indicating that this category is treated as a negative sample, thereby causing the $p_j$ to decrease. Conversely, when the $p_j$ is less than $q_j$, the gradient is negative, indicating that this category is treated as a positive sample, thereby causing the $p_j$ to increase. Therefore, the role of the loss $L_{CR}$ is to align the predicted probability $p_j$ with the target probability $q_j$.

Next, we analyze the gradient of the second term in the SCE loss function, referring to it as $L_{topk} = p_j \left( \sum_{k=1}^{K} p_k \log q_k - \log q_j \right)$.

Let $q_{\min}$ be the minimum value in $q_k$ and $q_{\max}$ be the maximum value in $q_k$. Since the sum of $p_i$ is 1,

$$\log q_{\min} = \sum_{k=1}^{K} p_k \log q_{\min} \leq \sum_{k=1}^{K} p_k \log q_k \leq \sum_{k=1}^{K} p_k \log q_{\max} = \log q_{\max}. \tag{A21}$$

Therefore, there exists $q_{zero}$ such that,

$$\sum_{k=1}^{K} p_k \log q_k - \log q_{zero} = 0, \tag{A22}$$

and when $q_j < q_{zero}$, the above term is positive, and when $q_j > q_{zero}$, the above term is negative.

$$\frac{\partial L_{topk}}{\partial z_j} = p_j \left( \sum_{k=1}^{K} p_k \log q_k - \log q_j \right) = \begin{cases} \leq 0, & q_j \geq q_{zero} \\ > 0, & q_j < q_{zero} \end{cases} \tag{A23}$$

Table A1: Detailed runtime performance analysis with the ResNet-18 backbone on a single Titan XP GPU on the PACS dataset.

| Exp ID | Method | Runtime (s) |
|--------|--------|-------------|
| #1 | ERM (ResNet-18) | 11.15 |
| #2 | +SHOT (Liang et al., 2020) | 11.15 + 20.97 = 32.12 |
| #3 | +SHOTIM (Liang et al., 2020) | 11.15 + 20.73 = 31.88 |
| #4 | +TSD (Wang et al., 2023) | 11.15 + 19.73 = 30.88 |
| #5 | +PGM | 11.15 + 1.21 = 12.36 |
| #6 | +RST | 11.15 + 16.78 = 27.93 |
| #7 | +PGM + RST (PROGRAM) | 11.15 + 17.92 = 29.07 |
| #8 | +PGM + CE Loss | 11.15 + 17.54 = 28.69 |
| #9 | +PGM + KL Loss | 11.15 + 17.65 = 28.80 |

The $L_{topk}$ term increases the prediction probability of several high-probability predictions and decreases the prediction probability of several low-probability predictions. Consequently, $L_{topk}$ treats classes with high-probability predictions as positive samples while considering the remaining classes as negative samples.

Note that the gradient of Cross-Entropy (CE) $L_{CE}$ is:

$$\frac{\partial L_{CE}}{\partial z_j} = \begin{cases} p_j - 1 \le 0, & q_j = 1 \\ p_j > 0, & q_j = 0 \end{cases} \tag{A24}$$

Therefore, the gradient of our overall loss function is as follows:

$$\frac{\partial L_{RST}}{\partial z_j} = \begin{cases} \begin{cases} p_j - 1, & q_j = 1 \\ p_j, & q_j = 0 \end{cases}, & \textbf{reliable pseudo-label} \\ p_j - q_j + p_j \left( \sum_{k=1}^{K} p_k \log q_k - \log q_j \right), & \textbf{unreliable pseudo-label} \end{cases} \tag{A25}$$

When pseudo-labels generated by the PGM are reliable, we use the Cross-Entropy (CE) loss function, which treats the class with the unique maximum probability as the positive sample. This enables the loss function to converge quickly. When pseudo-labels are less reliable, we employ the Symmetric Cross Entropy (SCE) loss function. The first gradient term of the SCE loss, as shown in Eq. A20, brings the predictions of the model closer to the predictions of the PGM. This can be seen as consistency regularization. The gradient of the second term of the SCE loss, as shown in Eq. A23, will increase the prediction probability of several high-probability predictions and reduce the prediction probability of several low-probability predictions (categories with high predicted probabilities are likely to contain the correct labels). Therefore, the SCE loss is more robust to noise labels.

## B    LIMITATIONS

It is still under exploration that how to extend our proposed PROGRAM to widespread computer vision tasks. Similar to the previous TTA methods, PROGRAM is currently applied to classification task and not adapted to regression problems such as object detection and semantic segmentation. We are actively working on creating a unified TTA framework that can be applied across all vision tasks. Another problem is that it is hard to evaluate or estimate the model's behaviour in real application scenarios because the model is constantly updated with the test data. This may raise some ethical concerns in some sensitive applications. It's worth noting that the existing TTA methods, such as T3A, TAST, Tent, and TSD, also encounter similar challenges.

## C    MORE DETAILED RUNTIME ANALYSIS

We report the breakdown of PROGRAM's runtime in Table A1. Our PGM module exhibits a relatively short computational time (comparing #1 and #5), making it efficient for reliable pseudo-label

Table A2: Effect of batch sizes on OfficeHome dataset with ResNet-50 backbone. Average accuracy(%) is reported. The higher the better.

| Method | 8 | 16 | 32 | 64 | 128 |
|---|---|---|---|---|---|
| ERM (ResNet-50) | 67.13±0.99 | 67.13±0.99 | 67.13±0.99 | 67.13±0.99 | 67.13±0.99 |
| +Tent | 62.10±1.44 | 64.84±1.86 | 66.29±0.77 | 66.67±1.26 | 67.00±1.80 |
| +SHOT | 66.07±0.85 | 67.00±1.20 | 67.65±0.72 | 67.73±1.37 | 67.99±1.41 |
| +PL | 57.51±3.26 | 60.70±3.12 | 62.85±3.05 | 63.32±1.92 | 63.70±1.81 |
| +T3A | 68.60±0.90 | 68.55±1.23 | 68.26±0.84 | 68.35±0.72 | 68.47±1.26 |
| +TAST | 68.70±0.60 | 68.76±0.97 | 68.63±0.70 | 68.60±0.53 | 68.99±1.24 |
| +TAST-BN | 68.04±1.67 | 68.42±1.12 | 68.88±0.50 | 69.23±1.55 | 69.36±1.98 |
| +TSD | 67.59±1.63 | 68.02±1.00 | 68.43±0.74 | 68.72±0.96 | 69.12±1.11 |
| +PROGRAM | **68.80±1.33** | **68.93±1.25** | **69.05±0.66** | **69.58±1.46** | **69.76±1.29** |

Table A3: Effect of batch sizes on CIFAR-100C dataset with ResNet-50 backbone. Average error rate (%) is reported. The lower the better.

| Method | 8 | 16 | 32 | 64 | 128 |
|---|---|---|---|---|---|
| ERM (ResNet-50) | 60.35 | 60.35 | 60.35 | 60.35 | 60.35 |
| +Tent | 45.56 | 43.25 | 41.54 | 40.16 | 39.04 |
| +SHOT | 47.48 | 45.13 | 43.71 | 42.27 | 41.54 |
| +PL | 46.18 | 44.78 | 42.13 | 41.48 | 40.06 |
| +T3A | 59.89 | 59.03 | 58.75 | 58.32 | 58.28 |
| +TAST | 64.12 | 62.63 | 61.54 | 60.96 | 60.74 |
| +TAST-BN | 43.45 | 41.87 | 39.12 | 38.48 | 37.82 |
| +TIPI | 39.36 | 39.14 | 38.86 | 38.65 | 38.33 |
| +TSD | 39.61 | 39.20 | 38.74 | 38.16 | 37.67 |
| +PROGRAM | **38.97** | **38.63** | **37.95** | **37.28** | **36.42** |

generation. And the main time consumption occurs during the back-propagation phase in RST. However, the computational time required for the RST module is comparable to other self-training modules that employ different loss functions (compare #7 with #8 and #9). Overall, our method demonstrates smaller computational costs compared to TTA methods such as SHOT (#2), SHOTIM (#3), and TSD (#4) that involve adjusting the whole model parameters. This highlights the practicality and feasibility of incorporating our method into real-world applications, as it achieves significant performance gains while maintaining reasonable computational efficiency.

# D  MORE EXPERIMENTAL ANALYSIS

## D.1  EFFECT OF BATCH SIZE.

We have incorporated OfficeHome (Venkateswara et al., 2017) and CIFAR-100C (Hendrycks & Dietterich, 2019) benchmarks into our evaluation to assess the impact of batch size. The results are in Table A2 and Table A3, respectively. We observe that our method consistently outperforms other approaches across various batch sizes. Moreover, as the batch size increases, the superiority of our proposed PROGRAM becomes even more pronounced, highlighting the advantages of our approach.

## D.2  ACCURACY OF PSEUDO-LABELS GENERATED BY PGM.

To further demonstrate the effectiveness of PGM in generating more reliable pseudo-labels, we conducted an additional experiment. The quality of pseudo-labels generated by PGM was explicitly evaluated in Table A4. Previous TTA methods commonly utilize the pseudo-labels generated by ERM directly (#1). The accuracy of the generated pseudo-labels by PGM is reported in #2. By comparing results #1 and #2, we demonstrate that PGM can produce superior pseudo-labels compared to the original ERM (68.90 vs. 68.25). We then leverage these generated pseudo-labels for robust self-training, resulting in the improved performance of model #3 (71.46). And in #4, we report the results obtained by applying PGM to generate the next round of pseudo-labels after fine-tuning. Consistent improvement is observed (71.98 vs. 71.46), highlighting the potential for iterative

Table A4: Effect of PGM in generating more reliable pseudo-labels. All experiments are conducted based on the ResNet-50 backbone. *Average accuracy* (%) is reported on domain generalization benchmarks. ↑ means higher is better.

| Exp ID | Method | VLCS ↑ | PACS ↑ | OfficeHome ↑ | TerraIncognita ↑ | Avg. ↑ |
|--------|--------|--------|--------|--------------|------------------|--------|
| #1 | ERM (ResNet-50) | 76.71±0.50 | 83.21±1.14 | 67.13±0.99 | 45.93±1.34 | 68.25 |
| #2 | +PGM (before fine-tune) | 77.36±0.61 | 84.16±0.93 | 67.69±1.35 | 46.38±1.79 | 68.90 |
| #3 | +PROGRAM | 78.17±0.92 | 90.78±0.40 | 69.05±0.66 | 47.84±1.23 | 71.46 |
| #4 | +PGM (after fine-tune) | 78.73±1.27 | 91.30±0.63 | 69.57±0.88 | 48.31±1.42 | 71.98 |

Table A5: Ablation study of PGM. All experiments are conducted based on the ResNet-50 backbone. *Average accuracy* (%) is reported on domain generalization benchmarks. ↑ means higher is better.

| Exp ID | Method | VLCS ↑ | PACS ↑ | OfficeHome ↑ | TerraIncognita ↑ | Avg. ↑ |
|--------|--------|--------|--------|--------------|------------------|--------|
| #1 | ERM (ResNet-50) | 76.71±0.50 | 83.21±1.14 | 67.13±0.99 | 45.93±1.34 | 68.25 |
| #2 | +PGM (with dot-product) + RST | 77.92±1.43 | 90.15±0.82 | 68.62±0.91 | 47.25±1.54 | 70.98 |
| #3 | +Non-parametric Attention + RST | 77.68±1.51 | 89.85±1.22 | 68.29±1.35 | 47.12±2.03 | 70.73 |
| #4 | +PROGRAM | **78.17±0.92** | **90.78±0.40** | **69.05±0.66** | **47.84±1.23** | **71.46** |

refinement. However, to ensure fair comparisons with other TTA methods, we do not report the results of #4 or the outcome of iterative refinement. These findings demonstrate that our method can produce more reliable pseudo-labels, leading to enhanced performance when compared with other TTA methods.

## D.3    ABLATION STUDY OF PGM

Prototype-based graph construction is based on the idea that we can use a small number of prototypes to turn sample-to-sample affinity computations into much simpler sample-to-prototype affinity computations. In addition, as prototypes contain global class-level information, they provide more robust and stable representations. This helps to mitigate the impact of noisy test samples. In Table A5, we conducted an ablative analysis for PGM.

**Comparisons with dot-product affinity computation.**    We first compare our sample-to-prototype affinity computations with the sample-to-sample affinity computations by dot-product. In Table A5, comparing #2 and #4, we observe that dot-product produces inferior performance. The reason for this discrepancy could be that the prototypes represented by linear layers and the test features may reside in different feature spaces and not follow the same distribution, leading to suboptimal performance when using dot-product as the affinity measure. In contrast, our proposed probability-based graph construction circumvents this issue, resulting in better performance. These results provide clear validation of the effectiveness of our proposed prototype-based graph construction and label propagation.

**Comparisons with non-parametric attention modules.**    We further compare our PGM with the non-parametric attention modules #3 in Table A5. Given the predefined graph, the non-parametric attention module first updates each node in the graph with the help of the adjacency matrix, and then use the updated class prototypes to make predictions of each updated test feature. The attention module have difficulty in efficiently exchanging message between prototypes represented by linear layers and the test features that may reside in different feature spaces and not follow the same distribution, which lead to inferior performance. In contrast, our proposed probability-based graph construction circumvents this issue, resulting in better performance. These findings demonstrate that our PGM outperforms the attention module alternative (71.46% vs 70.73%), confirming the effectiveness of our proposed prototype-based graph construction.

## D.4    ABLATION STUDY OF RST

We have conducted additional ablation studies to validate the effectiveness of our RST module in Table A6. We compare our RST approach with different strategies used in various TTA approaches,

Table A6: Ablation study of RST. All experiments are conducted based on the ResNet-50 backbone. *Average accuracy* (%) is reported on domain generalization benchmarks. ↑ means higher is better.

| Exp ID | Method | VLCS ↑ | PACS ↑ | OfficeHome ↑ | TerraIncognita ↑ | Avg. ↑ |
|---|---|---|---|---|---|---|
| #1 | ERM (ResNet-50) | 76.71±0.50 | 83.21±1.14 | 67.13±0.99 | 45.93±1.34 | 68.25 |
| #2 | +PGM + CE Loss (Wang et al., 2021b; Liang et al., 2020) | 77.72±0.83 | 90.45±0.37 | 68.23±0.78 | 46.92±1.04 | 70.83 |
| #3 | +PGM + CE Loss with confidence threshold=0.9 (Lee et al., 2013) | 77.76±1.26 | 90.56±0.76 | 68.28±0.89 | 47.01±0.92 | 70.90 |
| #4 | +PGM + CE Loss with M-th largest entropy (Iwasawa & Matsuo, 2021) | 77.77±1.71 | 90.60±1.13 | 68.32±0.94 | 47.03±1.28 | 70.93 |
| #5 | +PGM + KL Loss | 77.61±1.52 | 89.69±0.95 | 68.96±1.54 | 47.67±1.28 | 70.98 |
| #6 | +PGM + SCE Loss | 77.67±1.63 | 89.72±1.01 | 69.00±1.63 | 47.80±0.77 | 71.05 |
| #7 | +PGM + RST (PROGRAM) | 78.17±0.92 | 90.78±0.40 | 69.05±0.66 | 47.84±1.23 | 71.46 |

Table A7: Ablation study of normalized weights. All experiments are conducted based on the ResNet-50 backbone. *Average accuracy* (%) is reported on domain generalization benchmarks. ↑ means higher is better.

| Exp ID | Method | VLCS ↑ | PACS ↑ | OfficeHome ↑ | TerraIncognita ↑ | Avg. ↑ |
|---|---|---|---|---|---|---|
| #1 | ERM (ResNet-50) | 76.71±0.50 | 83.21±1.14 | 67.13±0.99 | 45.93±1.34 | 68.25 |
| #2 | +PGM (with unnormalized weights) + RST | 77.85±1.26 | 90.22±1.47 | 68.92±0.91 | 47.76±1.60 | 71.19 |
| #3 | +PGM (with normalized weights) + RST | 78.17±0.92 | 90.78±0.40 | 69.05±0.66 | 47.84±1.23 | 71.46 |

Table A8: Effect of partial updates. All experiments are conducted based on the ResNet-18 backbone. *Average accuracy* (%) is reported on domain generalization benchmarks. ↑ means higher is better.

| Exp ID | Method | VLCS ↑ | PACS ↑ | OfficeHome ↑ | TerraIncognita ↑ | Avg. ↑ | Runtime (s) |
|---|---|---|---|---|---|---|---|
| #1 | ERM (ResNet-18) | 76.71±0.50 | 83.21±1.14 | 67.13±0.99 | 45.93±1.34 | 68.25 | 11.15 |
| #2 | +PROGRAM (fine-tune linear) | 77.52±0.86 | 82.27±1.09 | 63.88±1.00 | 42.73±1.75 | 66.60 | 13.05 |
| #3 | +PROGRAM (fine-tune all) | 77.75±1.37 | 88.03±0.74 | 64.59±0.85 | 43.16±1.12 | 68.38 | 29.07 |

such as Tent (Wang et al., 2021b), SHOT (Liang et al., 2020), PL (Lee et al., 2013), and T3A (Iwasawa & Matsuo, 2021). The results are shown below. All experiments are conducted based on the ResNet-50 backbone. #1 is the baseline ERM. #2 replaces our RST module with the CE loss function, as done in Tent (Wang et al., 2021b) and SHOT (Liang et al., 2020). #3 improves upon #2 by introducing a confidence threshold=0.9, as done in PL (Lee et al., 2013). #4 further improves upon #2 by selecting only the pseudo-labels with lower Shannon entropy, filtering out 20% of low-quality data, as done in T3A (Iwasawa & Matsuo, 2021). #5 applies only the KL loss, which is commonly used for consistency regularization. #6 applies only the SCE (symmetric cross entropy) loss. Comparing all these alternatives, our RST module (#7) achieves the best results. Comparing all these alternatives, our RST module achieves the best results.

## D.5   EFFECT OF NORMALIZATION IN COMPUTING SOFT LABELS

In Eq. 1, the soft labels are estimated with the normalized prototypes. This is because the magnitudes of prototypes may vary. Without normalization, prototypes with larger magnitudes will exert a stronger influence on the computed value, which is not desired. We have conducted additional experiments to compare the results with and without normalization of prototypes. The results are shown in Table A7. The experiments clearly demonstrate the necessity of using normalized prototypes (#3) instead of unnormalized ones (#2).

## D.6   PARTIAL-UPDATE VARIANTS OF PROGRAM

We have conducted additional experiments to further analyze the computational efficiency of PROGRAM and explore the trade-off between effectiveness and efficiency. In Table A8, we investigated alternative fine-tuning approaches. Specifically, we design a partial-update variant of PROGRAM that only fine-tune the linear classification layer (#2). Based on the experimental results, fine-tuning only the linear layer shows promise and achieves a good trade-off between accuracy (66.60%) and efficiency (13.05s). However, it may perform suboptimally on datasets with large domain gaps, such as PACS. In order to achieve a model that performs well across all benchmark datasets, we choose to fine-tune all parameters in our default setting (#3).

# E  DISCUSSION OF RELATED WORKS

Table A9: Adaptation settings differ by their data and therefore losses during training and testing. We denote $x$ for data and $y$ for labels. Source and target domain are denoted with subscript $s$ and $t$.

| Setting | Source Data | Target Data | Train Loss | Test Loss |
|---|---|---|---|---|
| Fine-tuning | - | $x_t, y_t$ | $L(x_t, y_t)$ | - |
| Unsupervised Domain Adaptation | $x_s, y_s$ | $x_t$ | $L(x_s, y_s) + L(x_t)$ | - |
| Source-free Domain Adaptation | - | full data $x_t$ | - | $L(x_t)$ |
| Test-time Training | $x_s, y_s$ | $x_t$ | $L(x_s, y_s) + L(x_s)$ | $L(x_t)$ |
| Test-time Adaptation | - | batch data $x_t$ | - | $L(x_t)$ |

Domain adaptation addresses model generalization from source domain to target domain. Table A9 summarizes and compare different adaptation settings in terms of source and target data required and types of losses.

Fine-tuning (Donahue et al., 2014; Yosinski et al., 2014) directly use labeled target data to retrain the pre-trained model using a standard supervised loss. Unsupervised domain adaptation (UDA) (Pei et al., 2018; Saito et al., 2018; Zhang et al., 2020; Lin et al., 2022) seeks to use both the labeled source datasets and an unlabeled target dataset to align features across domains. Moreover, the adaption is achieved during the training stage. This assumes that the labeled data from the source domain and unlabeled data from the target domain can be accessible during training, which is not a practical setting for many applications. Source-free domain adaptation (SFDA) (You et al., 2021b; Wang et al., 2021b; Yan et al., 2021; Liang et al., 2020; Morerio et al., 2020; Kurmi et al., 2021; Zhou et al., 2022; Tang et al., 2021) aims to resolve the domain adaptation problem when the source data are not available. In this setup, the model is adapted using the unlabeled data solely from the target domain. However, this adaptation is usually made in an offline manner, requiring all target test data to be available during the adaptation process. In addition, it normally requires multiple epochs to train the model, which hinders the deployment on online testing scenarios. Test-time training (TTT) methods (Gidaris et al., 2018; Liu et al., 2021; Sun et al., 2020b) also optimize during testing, but alters the training stage to include additional self-supervised loss. We show that our test-time adaptation (TTA) setting is different from these settings. TTA only requires the pre-trained model and online unlabeled target data for adaptation during the inference stage.

More recently, Zhu & Koniusz (2023) propose the prototype-based label propagation to solve the problem of transductive few-shot learning, which is also related to our work. However, there are some distinguishing aspects to be highlighted. (1) Different research topics and settings: Zhu & Koniusz (2023) focus on transductive few-shot learning, where *labeled* support samples and *all* unlabeled queries are available, assuming training and testing occur in the *same domain*. Our paper, on the other hand, addresses online test-time adaptation, which involves fine-tuning a pre-trained model using *a batch of* unlabeled test data from the *target domain*, *without accessing to any labeled data*. (2) Distinct graph construction: In (Zhu & Koniusz, 2023), prototypes are not part of the graph and are updated separately using gradient descent. In contrast, we treat prototypes as graph vertices in our approach. (3) Varied prototype definitions: Zhu & Koniusz (2023) initialize prototypes as mean vectors of labeled support samples since labeled data is available. In contrast, we initialize prototypes using the model weights of the classification layer. (4) Offline vs. online approach: Zhu & Koniusz (2023) require access to all unlabeled queries and performs iterative prototype updates in an offline manner. In contrast, in our online test-time adaptation, the unlabeled test set arrives in an online manner. Our re-initialization design ensures that our prototypes stay up to date with our model and maintain their representation of global class characteristics. By highlighting these differences, we emphasize the unique contributions and focus of our work compared to Zhu & Koniusz (2023).

## F    EXPERIMENTAL SETTINGS

### F.1    DATASETS.

**PACS** (Li et al., 2017) contains 9,991 examples and 7 classes that are collected from 4 domains: art, cartoons, photos, and sketches. **OfficeHome** (Venkateswara et al., 2017) consists of 4 domains: art, clipart, product, and real, and includes 15,588 images and 65 classes. **VLCS** (Torralba & Efros, 2011) comprises four domains: Caltech101, LabelMe, SUN09, and VOC2007, and includes 10,729 images and 5 classes. **TerraIncognita** (Beery et al., 2018) is composed of wild animal images taken from 4 different locations (L100, L38, L43, and L46), and consists of 24,788 examples of 10 classes. **CIFAR-10C/100C** and **ImageNet-C** (Hendrycks & Dietterich, 2019) is constructed by corrupting the original clean CIFAR-10/100 (Krizhevsky, 2009) and ImageNet (Deng et al., 2009; Krizhevsky et al., 2012) datasets. The corruption consists of 15 different types, *i.e.*, Gaussian noise, shot noise, impulse noise, defocus blur, glass blue, motion blur, zoom blur, snow, frost, fog, brightness, contrast, elastic transformation, pixelation, and JPEG compression, in which each corruption type has 5 different severity levels and the larger severity level means more severe distribution shift. In the experiments, we mainly focus on the most severe corruption level.

### F.2    MODELS.

In the main experiments, we compared different methods on ResNet-18/50 (He et al., 2016), which are mostly widely used in domain adaptation and generalization community. To validate the generalization ability of the proposed method, we tested our method on different backbone architectures, including vision transformers (ViT-B/16 (Dosovitskiy et al., 2020)), hand-designed convolutional neural network (CNN) based models (ResNeXt-50 (Xie et al., 2017)), network architecture search (NAS) based models (EfficientNet-B4 (Niu et al., 2022)), and multi-layer perceptron (MLP) based models (Mixer-L/16 (Tolstikhin et al., 2021)). For various backbones, we use the `torchvision` implementation[1] except for ViT-B/16 and MLP-mixer, we use implementation from `timm` library[2].

### F.3    BASELINES

We compare PROGRAM with several baseline methods. For fair comparisons, we adapt these methods to fit in our online test-time adaptation setting. **Empirical Risk Minimization (ERM)** (Chowdhary, 2020) represents the standard supervised training baseline. **Pseudo Labeling (PL)** (Lee et al., 2013) fine-tunes the pre-trained classifier using confident pseudo-labels based on the model predictions. **PLClf** is a modified version of PL that fine-tunes only the last linear classifier. **Tent**[3] (Wang et al., 2021a) optimizes the model only the parameters of the BN layers by the minimize the entropy of its predictions. **TentAdapter**[3] (Wang et al., 2021a) is a modified version of Tent that adds a BN layer between the feature extractor and the last linear classifier, and optimizes only the added BN layer. **TentClf** [3] (Wang et al., 2021a) is a modified version of Tent that optimizes the model only the last linear classifier instead of the BN layers. **SHOTIM** [4] (Liang et al., 2020) fine-tunes the whole feature extractor to maximize the mutual information between test data and the model predictions. **SHOT** [4](Liang et al., 2020) is a modified version of SHOTIM that adds a pseudo-labeling loss. Originally, SHOT is one of the source-free domain adaptation methods which focus on the offline adaption setting. However, for a fair comparison, we compare with the online version of SHOT. **T3A**[5] (Iwasawa & Matsuo, 2021) predicts the labels of the test data by comparing distances between test data and the pseudo-prototypes generated by the linear classifier. **TAST**[6] (Jang et al., 2023) fine-tunes the pre-trained model via self-training with nearest neighbor information. **TAST-BN**[6] (Jang et al., 2023) is a modified version of TAST that adjusts the parameters of the BN layer. Note that the multiple projection layers of TAST can be regarded as a kind of ensemble learning, and increasing the number of projection layers can improve the accuracy. **TSD**[7] (Wang et al., 2023)

---

[1]https://github.com/pytorch/vision

[2]https://github.com/rwightman/pytorch-image-models

[3]https://github.com/DequanWang/tent

[4]https://github.com/tim-learn/SHOT

[5]https://github.com/matsuolab/T3A

[6]https://github.com/mingukjang/TAST

[7]https://github.com/SakurajimaMaiii/TSD

uses test time self-distillation to make the target feature as uniform as possible and uses memorized spatial local clustering to encourage the distance of feature representations to be aligned with the pseudo logits. **TIPI**[8] (Nguyen et al., 2023) uses transformation invariance as an unsupervised surrogate loss function for online domain adaptation.

## F.4 IMPLEMENTATION

### F.4.1 IMPLEMENTATION DETAILS ON DOMAIN GENERALIZATION BENCHMARKS.

We follow the same implementations in TAST (Jang et al., 2023) and TSD (Wang et al., 2023). We splite 80% of data from the source domains for training and 20% for validation. All models are initialized with ImageNet-1K (Russakovsky et al., 2015) pre-trained weights. For ResNet-18/50, the same hyperparameters are applied for training and test-time adaptation as in TAST (Jang et al., 2023) and DomainBed[9] (Gulrajani & Lopez-Paz, 2020). We set the batch size as 32 for each source domain and the learning rate as 5e-5 with Adam optimizer. We set dropout probability and weight decay to 0. For ViT-B/16 (Dosovitskiy et al., 2020), ResNeXt-50 (Xie et al., 2017), EfficientNet-B4 (Niu et al., 2022), and Mixer-L/16 (Tolstikhin et al., 2021), we use the same hyperparameters as in TSD (Wang et al., 2023). We use the Adam optimizer with a learning rate of 5e-5, weight decay of 0, dropout rate of 0, and a batch size of 32. We train source model for 5k iterations. All images are resized to $224 \times 224$ and data augmentation is used in source domain training, which includes random cropping, horizontal flipping, color jittering, and intensity changing. During test-time adaptation, the batch size is 32 unless explicitly stated. At the testing phase, PL (Lee et al., 2013), Tent and SHOT (Liang et al., 2020) utilize the SGD optimizer with a learning rate of 0.001 and a momentum of 0.9. Tent (Wang et al., 2021a), TAST (Jang et al., 2023), and TAST-BN (Jang et al., 2023) use the Adam optimizer with a learning rate of 0.001. For TSD (Wang et al., 2023), official parameters are adopted, using the Adam optimizer with a learning rate of 5e-5, and for TIPI (Nguyen et al., 2023), the SGD optimizer is used with a learning rate of 0.0025, following official hyperparameters. The confidence threshold for PL and PLClf is set to 0.9. For PL, only the BN layers in the trained model are adjusted, as in TAST (Wang et al., 2021a). For TAST-BN, the size of the entire support set is limited to 150. In all experiments, we use four random seeds $\{0, 1, 2, 3\}$ and report the average results in the main text.

### F.4.2 IMPLEMENTATION DETAILS ON IMAGE CORRUPTION BENCHMARKS.

We use the same implementation introduced in TAST (Jang et al., 2023). ResNet-18/50 are trained for 1000 epochs, combining classification and instance discrimination tasks. The instance discrimination task weight is set at 0.1 to balance the two tasks. For image corruption datasets, we employ the same data augmentation methods as in TAST (Nguyen et al., 2023), including random cropping, horizontal flipping, color jittering, and intensity changing. The batch size during network training is fixed at 256. At test time, for the CIFAR-10/100C datasets, the batch size is set at 128, and for the ImageNet-C dataset, it is 64. At the testing phase, PL, Tent and SHOT use the SGD optimizer with a learning rate of 0.001 and a momentum of 0.9. The confidence threshold for both PL and PLClf is maintained at 0.9. In PL, only the BN layers in the trained model are adjusted. Tent, TAST, and TAST-BN use the Adam optimizer with a learning rate of 0.001. There are no restrictions on the size of the support set for TAST-BN. TSD (Wang et al., 2023) applies the Adam optimizer with a learning rate of 5e-5 following official setting. For TIPI (Nguyen et al., 2023), the official setting is applied, using the SGD optimizer with a learning rate of 0.0025.

---

[8]https://github.com/atuannguyen/TIPI
[9]https://github.com/facebookresearch/DomainBed

# G  APPLYING PROTOTYPE GRAPH MODEL (PGM)

In this section, we introduce how we apply Prototype Graph Model (PGM) to the existing pseudo-labeling based TTA methods.

## G.1  T3A + PGM

For T3A (Iwasawa & Matsuo, 2021), we directly replace its pseudo-label generation process ($\hat{y} = \arg\max q_\omega(Y = y_k|f_\theta(x))$.) with PGM. We input features obtained by the backbone into PGM to generate pseudo-labels: $\hat{y}_{\text{PGM}} = \text{PGM}(f_\theta(x))$. Note that for a fair comparison, we do not change the update process of the T3A pseudo-prototype set with pseudo-labels. Our PGM only changes the pseudo-label generation process, and the inference procession follows the standard T3A process. The prototypes in the PGM will be the same as prototypes in $\mathbb{S}_k$. The difference is highlighted with blue colors.

## G.2  TAST + PGM AND TAST-BN + PGM

For TAST and TAST-BN (Jang et al., 2023), we also use PGM to replace the pseudo-label generation stage in TAST, without changing other parts. The inference process remains the same as the standard TAST or TAST-BN. The difference is highlighted with blue color. The prototypes in the PGM will be the same as prototypes in $\mathbb{S}$.

---

**Algorithm A1** Algorithm of T3A

---

**Input**: Feature extractor $f_\theta$, the batch of input $\mathbb{B}$, and support sets $\mathbb{S}_k$ available at this point.
**Output**: Prediction for all $x \in \mathbb{B}$, where $x \sim P(X)$.

Step 1. Adjust the template for each class using the $\mathbb{B}$.
for $x \in \mathbb{B}$:
$\hat{y} = \arg\max q_\omega(Y = y_k \mid f_\theta(x))$
$\mathbb{S}_k = \mathbb{S}_k \cup \left\{ \frac{f_\theta(x)}{\|f_\theta(x)\|} \right\}$ for $y_k = \hat{y}$
end for
Filter support sets.
Step 2. Predict based on the distance between the adjusted template.
**Return**: $\arg\max_\gamma(Y = y_k|f_\theta(x))$ for all $x \in \mathbb{B}$

---

---

**Algorithm A2** Algorithm of T3A + PGM

---

**Input**: Feature extractor $f_\theta$, the batch of input $\mathbb{B}$, and support sets $\mathbb{S}_k$ available at this point.
**Output**: Prediction for all $x \in \mathbb{B}$, where $x \sim P(X)$.

Step 1. Adjust the template for each class using the $\mathbb{B}$.
$\hat{y} = \text{PGM}(f_\theta(x))$
$\mathbb{S}_k = \mathbb{S}_k \cup \left\{ \frac{f_\theta(x)}{\|f_\theta(x)\|} \right\}$ for $y_k = \hat{y}$
Filter support sets.
Step 2. Predict based on the distance between the adjusted template.
**Return**: $\arg\max_\gamma(Y = y_k|f_\theta(x))$ for all $x \in \mathbb{B}$

---

---

**Algorithm A3** Algorithm of TAST

---

**Input**: Feature extractor $f_\theta$, number of adaptation modules $N_e$, adaptation modules $\{h_{\phi_i}\}_{i=1}^{N_e}$, test batch $\mathbb{B}$, support set $\mathbb{S}$, number of gradient steps per adaptation $T$, number of support examples per each class $M$, number of nearby support examples $N_s$, learning rate $\alpha$

**Output**: Predictions $\hat{y}_x$ for all $x \in \mathbb{B}$.

Step 1. Update the support set $\mathbb{S}$.

    for $t = 1 : T$ do

      for $i = 1 : N_e$ do

        for $x \in \mathbb{B}$ do

          $\hat{p}_{\text{TAST},i}(\cdot \mid x) =$ Obtain the nearest neighbor-based pseudo label of $x$

          $p_{\text{proto},i}(\cdot \mid x) =$ Compute the prototype-based class distribution of $x$

        end for

        $\phi_i \leftarrow \phi_i - \alpha \nabla \phi_i \frac{1}{|\mathbb{B}|} \sum_{x \in \mathbb{B}} \text{CE}(\hat{p}_{\text{TAST},i}(\cdot \mid x), p_{\text{proto},i}(\cdot \mid x))$

      end for

    end for

Step 2. Compute the predictions $\hat{y}_x$

---

**Algorithm A4** Algorithm of TAST + PGM

---

**Input**: Feature extractor $f_\theta$, number of adaptation modules $N_e$, adaptation modules $\{h_{\phi_i}\}_{i=1}^{N_e}$, test batch $\mathbb{B}$, support set $\mathbb{S}$, number of gradient steps per adaptation $T$, number of support examples per each class $M$, number of nearby support examples $N_s$, learning rate $\alpha$

**Output**: Predictions $\hat{y}_x$ for all $x \in \mathbb{B}$.

Step 1. Update the support set $\mathbb{S}$.

    for $t = 1 : T$ do

      for $i = 1 : N_e$ do

        for $x \in \mathbb{B}$ do

          $\hat{p}_{\text{TAST},i}(\cdot \mid x) = \text{PGM}(f_\theta(x))$

          $p_{\text{proto},i}(\cdot \mid x) =$ Compute the prototype-based class distribution of $x$

        end for

        $\phi_i \leftarrow \phi_i - \alpha \nabla \phi_i \frac{1}{|\mathbb{B}|} \sum_{x \in \mathbb{B}} \text{CE}(\hat{p}_{\text{TAST},i}(\cdot \mid x), p_{\text{proto},i}(\cdot \mid x))$

      end for

    end for

Step 2. Compute the predictions $\hat{y}_x$

---

# H FULL RESULTS

Table A10: Full results on the VLCS dataset with different TTA methods. * denotes the results from (Wang et al., 2023).

| Method | Backbone | C | L | S | V | Avg. |
|---|---|---|---|---|---|---|
| ERM* | ResNet-18 | 94.70±1.33 | 63.79±1.30 | 67.90±1.97 | 73.15±1.37 | 74.88 |
| +Tent* | ResNet-18 | 89.82±2.89 | 61.98±1.10 | 65.51±1.91 | 74.21±1.61 | 72.88 |
| +TentAdapter* | ResNet-18 | 79.80±4.74 | 58.51±1.44 | 61.62±0.92 | 68.14±1.74 | 67.02 |
| +TentClf* | ResNet-18 | 94.75±1.43 | 63.74±1.41 | 67.92±2.22 | 65.40±6.91 | 72.96 |
| +SHOT* | ResNet-18 | 91.45±6.83 | 48.26±1.77 | 54.75±2.59 | 66.51±1.25 | 65.24 |
| +SHOTIM* | ResNet-18 | 90.28±7.00 | 47.96±1.45 | 54.66±2.47 | 66.52±1.19 | 64.86 |
| +PL* | ResNet-18 | 93.57±2.24 | 53.82±2.51 | 50.58±9.50 | 53.91±2.78 | 62.97 |
| +PLClf* | ResNet-18 | 94.67±1.38 | 63.64±1.31 | 67.90±2.21 | 73.34±1.00 | 74.89 |
| +T3A* | ResNet-18 | 97.52±1.99 | 65.32±2.24 | 70.70±3.48 | 75.51±1.75 | 77.26 |
| +TAST* | ResNet-18 | 99.17±0.60 | 65.87±1.90 | 68.13±1.76 | 75.92±1.75 | 77.27 |
| +TAST-BN* | ResNet-18 | 92.60±8.66 | 64.75±1.29 | 67.27±3.14 | 76.23±3.73 | 75.21 |
| +TSD | ResNet-18 | 97.24±2.01 | 64.42±1.73 | 65.57±1.38 | 67.05±2.20 | 73.57 |
| +PROGRAM | ResNet-18 | 98.72±2.96 | 66.37±1.48 | 71.85±2.62 | 74.06±1.97 | 77.75 |
| +ERM* | ResNet-50 | 97.66±0.64 | 63.87±1.71 | 71.21±1.52 | 74.09±2.06 | 76.71 |
| +Tent* | ResNet-50 | 92.36±2.44 | 58.46±3.29 | 67.84±2.03 | 73.19±2.68 | 72.96 |
| +TentAdapter* | ResNet-50 | 85.36±3.49 | 58.35±3.46 | 66.47±2.71 | 68.42±2.11 | 69.65 |
| +TentClf* | ResNet-50 | 97.61±0.58 | 63.67±2.10 | 68.77±1.27 | 73.16±1.31 | 75.80 |
| +SHOT* | ResNet-50 | 98.72±1.50 | 46.82±2.57 | 55.70±1.78 | 67.04±2.88 | 67.07 |
| +SHOTIM* | ResNet-50 | 98.65±1.46 | 46.54±2.32 | 55.81±2.32 | 66.73±2.82 | 66.93 |
| +PL* | ResNet-50 | 98.48±0.34 | 53.45±2.82 | 59.45±9.24 | 66.24±8.63 | 69.41 |
| +PLClf* | ResNet-50 | 97.63±0.64 | 63.36±2.10 | 69.74±0.78 | 71.86±4.53 | 75.65 |
| +T3A* | ResNet-50 | 99.17±0.38 | 64.78±1.61 | 73.01±3.24 | 72.20±2.84 | 77.29 |
| +TAST* | ResNet-50 | 99.35±0.30 | 65.64±1.78 | 73.63±3.58 | 72.01±2.68 | 77.66 |
| +TAST-BN* | ResNet-50 | 96.09±2.40 | 60.22±6.08 | 65.78±6.51 | 71.99±5.90 | 73.52 |
| +TSD | ResNet-50 | 97.34±1.45 | 65.29±1.68 | 67.25±2.21 | 67.97±2.37 | 74.46 |
| +PROGRAM | ResNet-50 | 99.39±1.51 | 66.44±2.54 | 74.32±1.28 | 72.54±2.57 | 78.17 |

Table A11: Full results on the PACS dataset with different TTA methods. * denotes the results from (Wang et al., 2023).

| Method | Backbone | A | C | P | S | Avg. |
|---|---|---|---|---|---|---|
| ERM* | ResNet-18 | 77.78±0.81 | 75.09±1.22 | 95.19±0.29 | 69.11±1.22 | 79.29 |
| +Tent* | ResNet-18 | 82.21±1.07 | 81.20±0.51 | 95.32±0.33 | 76.82±1.97 | 83.89 |
| +TentAdapter* | ResNet-18 | 78.89±0.67 | 77.45±0.82 | 95.77±0.40 | 70.89±2.75 | 80.75 |
| +TentClf* | ResNet-18 | 78.16±1.05 | 75.01±1.53 | 95.50±0.35 | 65.60±5.96 | 78.57 |
| +SHOT* | ResNet-18 | 81.09±0.86 | 79.68±0.91 | 96.18±0.27 | 72.48±2.04 | 82.36 |
| +SHOTIM* | ResNet-18 | 81.10±0.90 | 79.66±0.95 | 96.18±0.27 | 72.35±2.03 | 82.33 |
| +PL* | ResNet-18 | 76.42±4.89 | 61.05±5.48 | 95.70±0.56 | 50.75±8.79 | 70.98 |
| +PLClf* | ResNet-18 | 79.09±1.41 | 75.46±2.93 | 95.43±0.32 | 62.48±7.31 | 78.11 |
| +T3A* | ResNet-18 | 78.81±0.97 | 77.14±1.20 | 95.92±0.36 | 71.44±1.63 | 80.83 |
| +TAST* | ResNet-18 | 80.56±0.53 | 78.26±0.99 | 96.44±0.20 | 72.52±0.77 | 81.94 |
| +TAST-BN* | ResNet-18 | 86.49±0.20 | 83.70±2.57 | 97.23±0.11 | 80.85±1.42 | 87.07 |
| +TSD | ResNet-18 | 86.81±1.26 | 85.04±1.14 | 94.39±0.86 | 82.00±1.03 | 87.06 |
| +PROGRAM | ResNet-18 | 87.05±1.17 | 85.77±0.92 | 97.15±0.72 | 82.16±1.23 | 88.03 |
| ERM* | ResNet-50 | 82.92±1.65 | 78.05±3.36 | 96.50±0.32 | 75.38±3.31 | 83.21 |
| +Tent* | ResNet-50 | 82.54±1.32 | 84.90±1.35 | 95.45±0.93 | 77.74±1.36 | 85.16 |
| +TentAdapter* | ResNet-50 | 82.75±2.01 | 79.50±2.26 | 96.78±0.20 | 75.73±3.22 | 83.69 |
| +TentClf* | ResNet-50 | 83.00±1.87 | 77.86±4.20 | 96.55±0.36 | 73.25±6.14 | 82.66 |
| +SHOT* | ResNet-50 | 84.67±1.70 | 80.17±1.39 | 96.58±0.52 | 74.86±2.95 | 84.07 |
| +SHOTIM* | ResNet-50 | 84.62±1.79 | 80.24±1.41 | 96.54±0.46 | 75.16±2.88 | 84.11 |
| +PL* | ResNet-50 | 84.59±5.51 | 76.35±2.57 | 96.41±0.68 | 69.54±11.22 | 81.72 |
| +PLClf* | ResNet-50 | 83.88±2.00 | 78.93±3.68 | 96.53±0.40 | 73.96±6.08 | 83.33 |
| +T3A* | ResNet-50 | 83.56±2.03 | 79.75±3.14 | 96.99±0.24 | 75.36±3.57 | 83.92 |
| +TAST* | ResNet-50 | 83.85±2.05 | 79.15±3.03 | 96.93±0.27 | 76.49±3.13 | 84.11 |
| +TAST-BN* | ResNet-50 | 87.11±2.04 | 88.50±1.93 | 97.79±0.47 | 83.23±1.42 | 89.16 |
| +TSD | ResNet-50 | 87.80±1.25 | 88.15±1.03 | 95.72±0.61 | 85.42±0.78 | 89.27 |
| +PROGRAM | ResNet-50 | 89.78±1.07 | 89.76±0.96 | 97.86±0.73 | 85.71±0.86 | 90.78 |

Table A12: Full results on the OfficeHome dataset with different TTA methods. * denotes the results from (Wang et al., 2023).

| Method | Backbone | A | C | P | R | Avg. |
|---|---|---|---|---|---|---|
| ERM* | ResNet-18 | 55.19±0.49 | 47.76±1.02 | 72.22±0.53 | 73.21±0.89 | 62.10 |
| +Tent* | ResNet-18 | 53.39±0.61 | 48.28±0.88 | 70.50±0.68 | 71.29±0.72 | 60.86 |
| +TentAdapter* | ResNet-18 | 55.53±0.43 | 49.53±0.95 | 72.47±0.27 | 73.01±1.23 | 62.64 |
| +TentClf* | ResNet-18 | 55.17±0.67 | 36.73±1.94 | 72.21±0.52 | 73.22±0.97 | 59.33 |
| +SHOT* | ResNet-18 | 55.14±0.57 | 50.27±1.18 | 71.69±0.45 | 73.21±0.91 | 62.58 |
| +SHOTIM* | ResNet-18 | 55.08±0.56 | 50.29±1.17 | 71.71±0.40 | 73.21±0.90 | 62.57 |
| +PL* | ResNet-18 | 54.49±1.06 | 34.66±13.13 | 71.45±0.37 | 72.20±0.65 | 58.20 |
| +PLClf* | ResNet-18 | 55.14±0.70 | 47.70±1.25 | 72.21±0.54 | 72.62±0.96 | 61.92 |
| +T3A* | ResNet-18 | 55.10±0.74 | 49.56±1.14 | 74.10±0.55 | 74.07±1.18 | 63.21 |
| +TAST* | ResNet-18 | 56.15±0.68 | 50.04±1.31 | 74.33±0.28 | 74.28±1.23 | 63.70 |
| +TAST-BN* | ResNet-18 | 55.11±0.58 | 51.35±0.85 | 72.58±0.80 | 72.13±0.78 | 62.79 |
| +TSD | ResNet-18 | 57.93±1.72 | 53.56±2.01 | 73.05±2.27 | 73.51±1.40 | 64.51 |
| +PROGRAM | ResNet-18 | 56.91±0.82 | 52.23±1.30 | 74.98±1.44 | 74.25±1.06 | 64.59 |
| ERM* | ResNet-50 | 61.32±0.69 | 53.44±1.11 | 75.84±1.10 | 77.90±0.92 | 67.13 |
| +Tent* | ResNet-50 | 60.98±0.67 | 53.94±1.24 | 74.49±0.71 | 75.75±0.53 | 66.29 |
| +TentAdapter* | ResNet-50 | 62.63±0.45 | 54.90±1.17 | 76.20±1.09 | 77.92±1.01 | 67.91 |
| +TentClf* | ResNet-50 | 61.35±0.73 | 52.72±1.40 | 75.23±1.05 | 77.86±1.07 | 66.79 |
| +SHOT* | ResNet-50 | 61.91±0.33 | 55.58±0.91 | 75.49±1.54 | 77.60±0.80 | 67.65 |
| +SHOTIM* | ResNet-50 | 61.84±0.32 | 55.63±0.92 | 75.56±1.60 | 77.57±0.79 | 67.65 |
| +PL* | ResNet-50 | 59.42±1.55 | 42.40±12.31 | 73.80±2.26 | 75.77±1.50 | 62.85 |
| +PLClf* | ResNet-50 | 61.35±0.40 | 52.87±1.96 | 75.86±1.09 | 77.94±1.10 | 67.01 |
| +T3A* | ResNet-50 | 61.91±0.59 | 55.07±1.14 | 77.39±1.38 | 78.67±0.61 | 68.26 |
| +TAST* | ResNet-50 | 62.43±0.80 | 55.81±1.26 | 77.46±1.07 | 78.83±0.93 | 68.63 |
| +TAST-BN* | ResNet-50 | 63.22±0.85 | 58.20±0.98 | 77.14±1.10 | 76.94±0.39 | 68.88 |
| +TSD | ResNet-50 | 61.60±1.40 | 57.25±0.79 | 78.21±1.31 | 76.67±0.76 | 68.43 |
| +PROGRAM | ResNet-50 | 62.96±0.85 | 57.31±1.58 | 77.87±1.02 | 78.05±0.49 | 69.05 |

Table A13: Full results on the TerraIncognita dataset with different TTA methods. * denotes the results from (Wang et al., 2023).

| Method | Backbone | L100 | L38 | L43 | L46 | Avg. |
|---|---|---|---|---|---|---|
| ERM* | ResNet-18 | 37.18±2.46 | 36.12±4.20 | 53.18±1.27 | 36.02±1.37 | 40.62 |
| +Tent* | ResNet-18 | 38.29±0.48 | 25.82±3.91 | 41.53±1.59 | 29.15±1.83 | 33.70 |
| +TentAdapter* | ResNet-18 | 40.55±1.46 | 37.44±2.22 | 46.33±1.32 | 35.30±1.26 | 39.91 |
| +TentClf* | ResNet-18 | 34.44±13.31 | 34.19±5.76 | 52.71±2.03 | 31.86±2.26 | 38.30 |
| +SHOT* | ResNet-18 | 33.87±0.66 | 28.58±2.10 | 40.99±2.07 | 30.83±1.26 | 33.57 |
| +SHOTIM* | ResNet-18 | 33.83±1.29 | 28.13±2.30 | 40.81±2.18 | 30.64±1.46 | 33.35 |
| +PL* | ResNet-18 | 51.92±1.19 | 35.61±20.74 | 39.97±10.98 | 22.26±8.21 | 37.44 |
| +PLClf* | ResNet-18 | 45.22±2.45 | 36.03±5.81 | 52.76±1.54 | 33.10±2.27 | 41.78 |
| +T3A* | ResNet-18 | 36.22±1.89 | 40.08±1.98 | 50.72±1.02 | 33.79±1.25 | 40.20 |
| +TAST* | ResNet-18 | 43.67±2.83 | 39.24±3.79 | 52.64±3.02 | 35.01±1.09 | 42.64 |
| +TAST-BN* | ResNet-18 | 51.06±7.31 | 32.74±7.54 | 41.70±2.86 | 32.21±3.05 | 39.43 |
| +TSD | ResNet-18 | 52.85±2.16 | 36.86±1.38 | 45.10±1.82 | 35.79±2.03 | 42.65 |
| +PROGRAM | ResNet-18 | 51.26±1.05 | 39.38±0.84 | 47.96±1.51 | 34.02±1.67 | 43.16 |
| ERM* | ResNet-50 | 46.84±1.96 | 43.24±2.51 | 53.32±1.92 | 40.30±1.93 | 45.93 |
| +Tent* | ResNet-50 | 41.20±2.71 | 29.72±3.59 | 41.35±2.92 | 36.03±2.85 | 37.08 |
| +TentAdapter | ResNet-50 | 46.64±1.17 | 41.11±3.16 | 49.31±1.05 | 38.52±2.04 | 43.89 |
| +TentClf* | ResNet-50 | 49.87±3.80 | 43.31±3.19 | 53.01±2.31 | 28.40±6.19 | 43.64 |
| +SHOT* | ResNet-50 | 36.17±2.70 | 29.80±2.92 | 41.00±0.30 | 33.83±1.86 | 35.20 |
| +SHOTIM* | ResNet-50 | 35.56±2.76 | 27.49±4.01 | 40.77±0.45 | 33.67±1.84 | 34.37 |
| +PL* | ResNet-50 | 56.75±5.78 | 46.12±1.03 | 29.44±10.14 | 20.06±4.65 | 38.09 |
| +PLClf* | ResNet-50 | 52.28±3.95 | 43.76±2.96 | 52.78±2.15 | 37.81±2.49 | 46.66 |
| +T3A* | ResNet-50 | 45.13±1.26 | 44.67±2.56 | 52.52±0.78 | 40.13±2.31 | 45.61 |
| +TAST* | ResNet-50 | 53.01±3.95 | 43.27±3.21 | 53.79±2.72 | 39.66±3.65 | 47.43 |
| +TAST-BN* | ResNet-50 | 55.75±2.37 | 33.92±9.86 | 43.87±4.70 | 32.33±4.40 | 41.47 |
| +TSD | ResNet-50 | 56.97±1.64 | 42.06±1.90 | 51.29±2.23 | 39.32±2.26 | 47.41 |
| +PROGRAM | ResNet-50 | 55.15±1.67 | 43.90±2.02 | 52.86±1.65 | 39.45±1.56 | 47.84 |

Table A14: Full results on the CIFAR-10C dataset with different TTA methods. * denotes the results from (Wang et al., 2023).

| Method | gauss | brit | contr | defoc | elast | fog | frost | glass | impul | jpeg | motn | pixel | shot | snow | zoom | Avg. |
|---|---|---|---|---|---|---|---|---|---|---|---|---|---|---|---|---|
| No Adaptation* | 48.73 | 7.01 | 13.27 | 11.84 | 23.38 | 29.41 | 28.24 | 50.78 | 57.00 | 19.46 | 23.38 | 47.88 | 44.00 | 21.93 | 10.84 | 29.14 |
| +SHOT* | 17.09 | 8.64 | 8.57 | 9.83 | 19.53 | 19.72 | 13.93 | 25.60 | 27.15 | 13.98 | 14.01 | 11.68 | 16.02 | 15.89 | 8.22 | 15.32 |
| +Tent* | 15.91 | 7.91 | 7.85 | 9.27 | 18.13 | 16.45 | 12.62 | 23.48 | 24.52 | 13.19 | 12.70 | 10.93 | 14.59 | 14.06 | 7.68 | 13.95 |
| +PL* | 33.56 | 7.54 | 11.53 | 10.60 | 20.21 | 23.86 | 21.78 | 38.36 | 43.64 | 16.88 | 18.72 | 29.83 | 30.43 | 18.75 | 9.43 | 22.34 |
| +T3A* | 41.87 | 7.30 | 13.61 | 11.99 | 22.06 | 28.52 | 27.13 | 44.10 | 54.26 | 18.71 | 22.54 | 37.53 | 37.84 | 21.97 | 10.72 | 26.68 |
| +TAST* | 42.02 | 7.34 | 13.55 | 11.86 | 21.38 | 28.58 | 26.51 | 44.99 | 54.19 | 18.96 | 22.55 | 37.08 | 37.62 | 21.84 | 10.64 | 26.61 |
| +TAST-BN* | 14.91 | 7.68 | 7.81 | 8.62 | 16.81 | 15.10 | 12.25 | 21.82 | 22.54 | 12.38 | 11.67 | 10.34 | 13.77 | 12.99 | 7.57 | 13.08 |
| +TIPI | 18.44 | 5.93 | 5.17 | 5.68 | 18.60 | 16.50 | 12.88 | 24.42 | 26.86 | 10.53 | 10.99 | 10.03 | 15.29 | 16.03 | 5.47 | 13.52 |
| +TSD | 15.69 | 6.39 | 8.89 | 7.08 | 17.64 | 15.63 | 11.56 | 23.67 | 22.79 | 12.17 | 9.00 | 10.04 | 15.70 | 10.89 | 8.62 | 13.05 |
| +PROGRAM | 13.22 | 5.56 | 5.48 | 6.34 | 15.29 | 13.91 | 11.75 | 21.42 | 22.80 | 11.94 | 10.82 | 9.72 | 12.83 | 10.91 | 6.61 | 11.91 |

Table A15: Full results on the CIFAR-100C dataset with different TTA methods. * denotes the results from (Wang et al., 2023).

| Method | gauss | brit | contr | defoc | elast | fog | frost | glass | impul | jpeg | motn | pixel | shot | snow | zoom | Avg. |
|---|---|---|---|---|---|---|---|---|---|---|---|---|---|---|---|---|
| No Adaptation* | 80.77 | 28.86 | 50.93 | 39.62 | 59.54 | 68.11 | 60.19 | 54.79 | 82.26 | 87.75 | 49.96 | 54.22 | 72.27 | 77.84 | 54.58 | 60.35 |
| +SHOT* | 45.95 | 30.14 | 31.93 | 32.81 | 46.19 | 49.49 | 40.65 | 54.79 | 57.02 | 37.99 | 39.22 | 37.57 | 44.33 | 44.08 | 30.97 | 41.54 |
| +Tent* | 43.02 | 29.65 | 30.52 | 31.48 | 43.88 | 44.03 | 39.21 | 50.91 | 53.10 | 36.22 | 36.31 | 34.10 | 41.58 | 41.85 | 29.73 | 39.04 |
| +PL* | 43.94 | 30.14 | 31.20 | 32.11 | 45.07 | 46.57 | 40.11 | 52.66 | 54.48 | 37.48 | 36.92 | 34.59 | 42.68 | 42.77 | 30.19 | 40.06 |
| +T3A* | 76.95 | 29.54 | 48.02 | 39.64 | 55.68 | 65.90 | 58.45 | 78.23 | 86.39 | 48.82 | 53.46 | 66.31 | 74.14 | 55.01 | 37.68 | 58.28 |
| +TAST* | 80.13 | 29.40 | 50.86 | 40.43 | 58.13 | 69.24 | 60.89 | 81.94 | 88.94 | 50.44 | 57.26 | 70.58 | 77.47 | 56.98 | 38.46 | 60.74 |
| +TAST-BN* | 42.01 | 29.00 | 30.20 | 30.74 | 42.97 | 41.02 | 38.19 | 48.95 | 51.20 | 35.70 | 35.03 | 33.38 | 40.01 | 39.88 | 29.00 | 37.82 |
| +TIPI | 41.91 | 27.68 | 30.71 | 30.69 | 43.17 | 42.98 | 39.98 | 48.30 | 50.61 | 37.04 | 34.54 | 34.22 | 40.18 | 43.13 | 29.78 | 38.33 |
| +TSD | 41.41 | 28.55 | 29.11 | 31.02 | 41.24 | 42.36 | 37.30 | 50.05 | 51.62 | 35.71 | 34.95 | 34.98 | 38.57 | 40.55 | 27.68 | 37.67 |
| +PROGRAM | 42.52 | 27.87 | 30.10 | 29.76 | 39.50 | 39.85 | 37.60 | 46.65 | 48.26 | 34.18 | 33.91 | 31.31 | 39.23 | 37.28 | 28.32 | 36.42 |

