# OpenReview forum: "PROGRAM: PROtotype GRAph Model based Pseudo-Label Learning for Test-Time Adaptation"
_ICLR.cc/2024/Conference — ICLR 2024 poster_

### Official Review · Reviewer_XKXi · 2023-10-24

**Soundness:** 3 good
**Presentation:** 4 excellent
**Contribution:** 3 good
**Rating:** 6
**Confidence:** 5

**Summary:**

This paper presented a novel test-time adaptation method named PROGRAM based on pseudo-label learning. The key motivation of PROGRAM was to leverage label propagation to improve the quality of pseudo-labels and then use robust self-training for model updating. Experiments on several data sets confirmed that PROGRAM was effective and computationally efficient for test-time adaptation.

**Strengths:**

**Originality:** The major contribution of this paper was to introduce a novel PROGRAM method for test-time adaptation, which aimed to adapt a pre-trained source model to the test data. PROGRAM addressed the issues of noisy pseudo-labels by introducing two novel components. One was the Prototype Graph Model (PGM) for generating high-quality pseudo-labels. The other one was the Robust Self-Training (RST) by combining pseudo-labeling and consistency regularization for model updating. Experiments demonstrated that PROGRAM could achieve better prediction accuracy and comparable computational efficiency compared to state-of-the-art TTA baselines.

**Quality:** The motivation of PROGRAM in handling noisy pseudo-labels was clearly illustrated. Previous works using noisy pseudo-labels might lead to sub-optimal solutions in model updates. The graph-based label propagation improved pseudo-labels by enforcing that two similar samples have similar pseudo-labels. Experiments also demonstrated that with improved pseudo-labels, PROGRAM achieved promising performance in test-time adaptation.

**Clarity:** The presentation of this paper is easy to follow. It illustrates the technical details and experimental settings in this paper. Algorithm 1 also clearly illustrates the training and inference process of PROGRAM in test-time adaptation scenarios.

**Significance:** PROGRAM improves the performance of test-time adaptation with respect to various pre-trained source models. Thus it can be a strong baseline in test-time adaptation, especially in understanding the impact of pseudo-labels.

**Weaknesses:**

W1: The improvement of PROGRAM on the pseudo-labels is not quantitively evaluated. One key intuition behind PROGRAM is to improve pseudo-labels with PGM and RST. Though the ablation studies validate the necessity of PGM and RST in improving prediction accuracy, it is more convincing to explicitly evaluate the impact of these components on the quality of pseudo-labels.

W2: The similarity $w_{ij}$ in constructing the prototype graph is not well explained. In Eq. (2), the similarity $w_{ij}$ is defined over $p(v_i | v_j)$. In this case, $p(v_i | v_j) = p(v_j | v_i)$ for guaranteeing the symmetric similarity matrix. Equivalently, it might hold that $p(v_i , v_j) = p(v_i | v_j) p(v_j) = p(v_j | v_i) p(v_i)$ and then $p(v_i) = p(v_j)$ for all nodes. It is unclear whether it assumes that all samples follow a uniform distribution. This might be a strong assumption in real scenarios.

W3: Another concern is the computational efficiency of PROGRAM. It requires the graph construction and the matrix inverse computation. Besides, it also updates the "whole" feature extractors for RST. Both strategies might significantly increase the running time of PROGRAM for test-time adaptation, compared to other partial updating methods. Thus the trade-off between the effectiveness and efficiency of PROGRAM can be further analyzed, e.g., sparse graph construction, partial parameter updates, etc.

**Questions:**

Q1: The Robust Self-Training (RST) combines both pseudo-labels and consistency regularization. The effectiveness of RST can be validated with more ablation studies. For example, compared to simple pseudo-labeling or consistency regularization, how can their combination perform better on the TTA benchmarks?

Q2: The soft labels in Eq. (1) are estimated with the normalized model weights $c_k$ (prototypes). Compared to vanilla soft labels with the raw source model (with unnormalized weights), can these prototypes help provide better pseudo-labels?

Q3: Is the matrix $\mathbb{I} - \lambda \mathbf{Z} \mathbf{D}^{-1} \mathbf{Z}^T$ full rank to directly compute the inverse of this matrix?

Overall, this paper introduced an interesting idea for handling the noisy labels within the test-time adaptation and achieved promising performance in several benchmarks. I would like to increase my rating if my concerns can be well addressed.


##########################################################################################

Most of my concerns are addressed after rebuttal, thus I would like to increase my score. More discussion can be provided in this paper to validate the high-quality pseudo-labels of the proposed framework. The efficiency of PROGRAM can also be improved when the graph-based method is used to improve pseudo-labels.

---

> ### Author Response · Authors · 2023-11-21
> **Response to Reviewer XKXi (1/4)**
>
> **Q1: The improvement of PROGRAM on the pseudo-labels is not quantitively evaluated.**
>
> We appreciate the reviewer's important question regarding the quality of pseudo-labels generated by PGM. We conducted an additional experiment to explicitly evaluate the accuracy of the PGM module. The results are shown below.
>
> Exp ID| Method            | VLCS      | PACS      | OfficeHome | TerraIncognita | Avg.   |
> |---|-------------------|-----------|-----------|------------|-----------------|--------|
> | #1 | ERM (ResNet-50)     |  76.71±0.50 | 83.21±1.14 | 67.13±0.99 | 45.93±1.34      | 68.25  |
> | #2 | ERM + PGM (before fine-tune)          |  77.36±0.61| 84.16±0.93 | 67.69±1.35 | 46.38±1.79      | 68.90  |
> | #3 | ERM + PROGRAM          |  78.17±0.92 | 90.78±0.40 | 69.05±0.66 | 47.84±1.23 | 71.46|
> | #4 | ERM + PGM (after fine-tune)         |  78.73±1.27 | 91.30±0.63 | 69.57±0.88 | 48.31±1.42 | 71.98|
>
>
> Previous TTA methods commonly utilize the pseudo-labels generated by ERM directly. By comparing results #1 and #2, we demonstrate that PGM can produce superior pseudo-labels compared to the original ERM (68.90 vs. 68.25). We then leverage these generated pseudo-labels for robust self-training, resulting in the improved performance of model #3 (71.46). In #4, we report the results obtained by applying PGM to generate the next round of pseudo-labels after fine-tuning. Consistent improvement is observed (71.98 vs. 71.46), highlighting the potential for iterative refinement. However, to ensure fair comparisons with other TTA methods, we do not report the results of #4 or the outcome of iterative refinement. These findings demonstrate that our method can produce more reliable pseudo-labels, leading to enhanced performance when compared with other TTA methods. We have included these results in the revised version.
>
>
>
> **Q2: It is unclear whether it assumes that all samples follow a uniform distribution. This might be a strong assumption in real scenarios.**
>
> We appreciate the reviewer's concern regarding the assumption of a uniform distribution. Indeed, this is the general assumption of TTA settings. TTA generally assumes that all samples in the test data come from the unique target domain and follow the same distribution. While this assumption may appear strong, it aligns with real-world applications. Transfering a source domain model to **one** target domain is very common, particularly for TTA. For example, the model might be pre-trained under controlled lab environments, but deployed to a certain camera installed at a certain position. As the camera view is fixed, the test data in this case can be assumed to come from the same domain. TTA is essential to enhance the pre-trained model's performance in **that** target domain and data distribution.

---

> ### Author Response · Authors · 2023-11-21
> **Response to Reviewer XKXi (2/4)**
>
> **Q3: Concerns about computational efficiency. The trade-off between the effectiveness and efficiency of PROGRAM can be further analyzed, e.g., sparse graph construction, partial parameter updates, etc**
>
> Thank you for your valuable suggestions. We have conducted additional experiments to further analyze the computational efficiency of PROGRAM and explore the trade-off between effectiveness and efficiency.
>
> Firstly, we provide a breakdown of PROGRAM’s runtime, as suggested by Reviewer C9fo and DVoa. Our PGM module exhibits a relatively short computational time (comparing #1 and #5), making it efficient for reliable pseudo-label generation. And the main time consumption occurs during the back-propagation phase in RST. However, the computational time required for the RST module is comparable to other self-training modules that employ different loss functions (compare #7 with #8 and #9). Overall, our method demonstrates smaller computational costs compared to TTA methods such as SHOT (#2), SHOTIM (#3), and TSD (#4) that involve adjusting the whole model parameters. This highlights the practicality and feasibility of incorporating our method into real-world applications, as it achieves significant performance gains while maintaining reasonable computational efficiency.
>
> **Absolute runtime performance analysis with ResNet-18 backbone on a single Titan XP GPU on the PACS dataset.**
> | Exp ID | Method                           | Runtime (s)           |
> | ------ | -------------------------------- | --------------------- |
> | Baseline  |                             |                       |
> | #1     | ERM (ResNet-18)                  | 11.15                 |
> | #2     | ERM + SHOT (Liang et al, 2020)   | 11.15 + 20.97 = 32.12 |
> | #3     | ERM + SHOTIM (Liang et al, 2020) | 11.15 + 20.73 = 31.88 |
> | #4     | ERM + TSD (Wang et al, 2023)     | 11.15 + 19.73 = 30.88 |
> | Ours   |                                  |                       |
> | #5     | ERM + PGM                        | 11.15 + 1.21 = 12.36  |
> | #6     | ERM + RST                        | 11.15 + 16.78 = 27.93 |
> | #7     | **ERM + PGM + RST (PROGRAM)**   | 11.15 + 17.92 = 29.07 |
> | #8     | ERM + PGM + CE Loss              | 11.15 + 17.54 = 28.69 |
> | #9     | ERM + PGM + KL Loss              | 11.15 + 17.65 = 28.80 |
>
>
>
>
> Additionally, following your suggestions, we have conducted additional experiments to balance accuracy and efficiency. The results are shown in the table below.
>
> 1. We introduce a sparse graph construction method (#2). As PGM itself is relatively lightweight, the modifications to PGM have minimal impact on time consumption (28.62s vs 29.07s).
> 2. We also investigate serveral partial parameter update approaches in the experiments. Specifically, we compared approaches that only fine-tune the linear classification layer (#3), only fine-tune the backbone (#4), and only fine-tune the Batch Normalization (BN) layer (#5). Based on the experimental results, fine-tuning only the linear layer achieves a good trade-off between accuracy (66.60%) and efficiency (13.05s), which could serve as an alternative according to different practical requirements. Compared with our original PROGRAM, it performs suboptimally on datasets with large domain gaps, such as PACS. Please refer to the response to Q3 of Reviewer DVoa for a more detailed explanation. Fine-tuning all the parameters is our default setting in order to achieve a model that performs well across all the scenarios.
>
> |Exp ID| Method            |  VLCS      | PACS      | OfficeHome | TerraIncognita | Avg.   |Runtime (s)|
> |---|-------------------|-----------|----------|------------|-----------------|--------|----|
> |#1| ERM (ResNet-18)               |  74.88±0.46 |79.29±0.77| 62.10±0.31| 40.62±1.19| 64.22  |11.15|
> |#2| +PROGRAM (sparse graph) |77.29±1.02  |87.74±1.45 |64.12±1.17  | 42.97±0.83 |68.03 |28.62|
> |#3| +PROGRAM (fine-tune linear)         | 77.52±0.86 | 82.27±1.09 | 63.88±1.00 | 42.73±1.75    | 66.60   |13.05|
> |#4| +PROGRAM (fine-tune backbone)         | 77.22±1.49 | 87.65±1.87 | 64.08±1.98 | 42.83±1.50   | 67.94  |28.99|
> |#5| +PROGRAM (fine-tune BN layer)         | 76.73±1.20 | 87.04±1.63 | 63.21±0.97 |  42.16±1.23 |67.28  |28.86 |
> |#6| +PROGRAM (our final version: fine-tune all)         | 77.75±1.37| 88.03±0.74| 64.59±0.85| 43.16±1.12| 68.38|29.07|

---

> ### Author Response · Authors · 2023-11-21
> **Response to Reviewer XKXi (3/4)**
>
> **Q4: The effectiveness of RST can be validated with more ablation studies.**
>
> Thank you for your valuable feedback. We have addressed your concerns by providing a theoretical analysis and conducting additional ablation studies to validate the effectiveness of our RST module.
>
> 1. **Theoretical Analysis**: We have included a detailed theoretical analysis in the Appendix (Sec. A3) to explain why our loss function is more effective. The analysis highlights how our RST module combines the advantages of two loss functions. When the generated pseudo-labels are reliable, we use the Cross-Entropy (CE) loss function, which treats the class with the maximum probability as the positive sample. This enables the self-training to converge quickly. When pseudo-labels are less reliable, we employ the Symmetric Cross Entropy (SCE) loss function. This analysis indicates that the SCE loss not only imposes consistency regularization but also simultaneously learns multiple categories with high predicted probabilities, thereby alleviating the impact of noisy pseudo-labels.
> 2. **Ablation Studies**: We have conducted additional ablation studies to validate the effectiveness of our RST module. We compare our RST approach with different strategies used in various TTA approaches, such as Tent, SHOT, PL, and T3A. The results are shown below. All experiments are conducted based on the ResNet-50 backbone. #1 is the baseline ERM. #2 replaces our RST module with the CE loss function, following Tent and SHOT. #3 improves upon #2 by introducing a confidence threshold=0.9, following PL. #4 modifies #2 by selecting the pseudo-labels with lower Shannon entropy, filtering out 20% of low-quality data, following T3A. #5 and #6 respectively apply KL loss and SCE loss for consistency regularization only. Comparing all these alternatives, our RST module (#7) achieves the best results.
>
>
> ExpID| Method         | Used By   |  VLCS      | PACS      | OfficeHome | TerraIncognita | Avg.   |
> |---|---------------|----|-----------|-----------|------------|-----------------|--------|
> | #1 | ERM (ResNet-50)   | -   |  76.71±0.50| 83.21±1.14| 67.13±0.99| 45.93±1.34| 68.25|
> | #2 | +CE Loss | Tent (Wang et al., 2021a) and SHOT (Liang et al., 2020) | 77.72±0.83 |90.45±0.37| 68.23±0.78 |46.92±1.04| 70.83|
> | #3 | +CE Loss with confidence threshold=0.9| PL (Lee et al., 2013) | 77.76±1.26 |90.56±0.76| 68.28±0.89 |47.01±0.92| 70.90|
> | #4 | +CE Loss with M-th largest entropy | T3A(Iwasawa & Matsuo, 2021)|77.77±1.71 |90.60±1.13| 68.32±0.94 |47.03±1.28| 70.93|
> | #5 | +KL Loss | - |  77.61±1.52| 89.69±0.95| 68.96±1.54| 47.67±1.28| 70.98|
> | #6 | +SCE Loss| - | 77.67±1.63| 89.72±1.01| 69.00±1.63| 47.80±0.77| 71.05|
> | #7 | **+RST (PROGRAM)**| Ours | 78.17±0.92| 90.78±0.40| 69.05±0.66| 47.84±1.23| 71.46|
>
> By providing a theoretical analysis and conducting additional ablation studies, we have thoroughly validated the effectiveness of our RST module.
>
>
> **Q5: Compared to vanilla soft labels with the raw source model (with unnormalized weights), can these prototypes help provide better pseudo-labels?**
>
> It is important to note that the magnitudes of prototypes may vary. Without normalization, prototypes with larger magnitudes will exert a stronger influence on the computed value, which is not desired. We have conducted additional experiments to compare the results with (#3) and without (#2) normalization of prototypes. The experiments clearly demonstrate the necessity of using normalized prototypes instead of unnormalized ones.
>
>
> |Exp ID| Method            | VLCS      | PACS      | OfficeHome | TerraIncognita | Avg.   |
> |-----------------|-------------------|-----------|-----------|------------|-----------------|--------|
> |#1| ERM (ResNet-50)  |  76.71±0.50 | 83.21±1.14 | 67.13±0.99 | 45.93±1.34      | 68.25  |
> |#2| +PGM (with unnormalized weights) + RST   |  77.85±1.26 | 90.22±1.47 | 68.92±0.91 | 47.76±1.60      |  71.19 |
> |#3| +PGM (with normalized weightst) + RST   |  78.17±0.92| 90.78±0.40| 69.05±0.66| 47.84±1.23| 71.46|

---

> ### Author Response · Authors · 2023-11-21
> **Response to Reviewer XKXi (4/4)**
>
> **Q6: Is the matrix $\bf{I} - \lambda ZD^{-1}Z^T$ full rank to directly compute the inverse of this matrix?**
>
> With some calculations (please refer to Sec. A2 for more details), we can get:
>
> $ |\mathbb{I} - \lambda \mathbf{Z}\mathbf{D}^{-1}\mathbf{Z}^T| = |\mathbf{B}_{K \times K}| \cdot |\mathbb{I} - \lambda (\mathbf{S} \mathbf{X} \mathbf{S}^T)| $
>
> where $ \mathbf{B} \in \mathbb{R}^{N\times N} $ is a diagonal matrix with all values greater than zero, ensuring $|\mathbf{B}_{K \times K}| > 0 $. Additionally, $ \mathbf{X} \in \mathbb{R}^{N\times N} $ is also a diagonal matrix with all values greater than zero. Considering that $S_i$ and $S_j$ represent class probability vectors of two test samples and generally exhibit no linear correlation, we expect the rank of $ \mathbb{I}-\lambda (\mathbf{S} \mathbf{X} \mathbf{ S}^T) $ to be $N$. Consequently, $ |\mathbb{I} - \lambda \mathbf{Z} \mathbf{D}^{-1} \mathbf{Z}^T| \neq 0 $, indicating the invertibility of $ \mathbb{I} - \lambda \mathbf{Z} \mathbf{D}^{-1} \mathbf{Z}^T $.
>
> Notably, in our experiments, we have not encountered non-invertible cases for this matrix. However, if a non-invertible situation arises, a small perturbation matrix $\mathbf{\Sigma} $ can be introduced to ensure invertibility: $(\mathbf{I} - \lambda \mathbf{Z} \mathbf{D}^{-1} \mathbf{Z}^T) + \epsilon \mathbf{\Sigma} $, where $ \epsilon $ is a positive number tending to zero. This regularization technique is commonly used, with $ \epsilon $ referred to as the regularization parameter.

---

> > ### Comment · Reviewer_XKXi · 2023-11-21
> > **Comments**
> >
> > Thanks for sharing the rebuttals. But I have several follow-up questions.
> >
> > (1) Regarding Q2, I agree that all samples in the test data come from the unique target domain, and thus they follow the same distribution. But my concern is why samples have to follow a **uniform** distribution. For example, for any target samples, it is reasonable to assume $x_1, x_2 \sim \mathbf{P}$ indicating both $x_1$ and $x_2$ follow the same distribution $\mathbf{P}$. But it does not imply that $x_1$ and $x_2$ must have identical sampling probability, i.e., $\mathbf{P}(x_1) = \mathbf{P}(x_2)$ (corresponding to the uniform distribution in my concern W2).
> >
> > (2) The proof of the inversion of $\mathbf{I} - \lambda \mathbf{ZD^{-1}Z^T}$ is not convincing.
> > - It is confusing why all values in $\mathbf{B}$ and $\mathbf{X}$ are greater than zero. The appendix lacks sufficient explanation. Is it also determined by the selected value of $\lambda$?
> > - It states that $S_i$ and $S_j$ represent class probability vectors of two test samples and generally exhibit no linear correlation. On one hand, it is a strong assumption that class probability vectors of two test samples have no linear correlation. If two samples are drawn from the same class, it has a high probability regarding the linear correlation between their class probability vectors. On the other hand, even though it is true that class probability vectors of any two test samples have no linear correlation, it is hard to follow how this can guarantee that the rank of $\mathbb{I} - \lambda \mathbf{SXS^T}$ is $N$.

---

> > > ### Author Response · Authors · 2023-11-22
> > > **Response to further comments.**
> > >
> > > Thanks for the timely response! We greatly appreciate your kind clarification on the concerns.
> > >
> > > **Q1. About uniform distribution.** Thanks for your clarification. In the context of graph construction with Markov random walk, there are N test samples in a batch, where N is the batch size. We traverse all of the test samples for graph construction, the sampling probability of all test samples are **uniform**, meaning p(x1) = p(x2) = ... = 1/N, where N is the batch size. This assuption is common and reasonable.
> > >
> > >
> > > **Q2. About the proof of matrix invertibility.**
> > >
> > > In our previous response, we intended to follow the proof outlined in paper[a] to demonstrate the invertibility of the matrix. The proof involves establishing the similarity of the matrix to a random matrix as a means to prove its invertibility. However, this proof is technically intricate and challenging to comprehend.
> > >
> > >
> > > Subsequently, we devised a more clever and elegant way to prove the invertibility of the matrix. We deliver the main idea here, and more details can be found in Sec. A.2.
> > >
> > >
> > > $(\bf{I} - \lambda \bf{W})$ can be rewritten as:
> > >
> > > \begin{align}
> > >     \bf{I} - \lambda \bf{W} = (1 - \lambda) \bf{I} + \lambda ( \bf{I} - \bf{W}).
> > > \end{align}
> > >
> > >
> > > As $\lambda = \frac{1}{1+\mu}$ is a number greater than 0 and less than 1, matrix $(1-\lambda) \bf{I}$ is a positive definite matrix. And the matrix $\bf{I}-\bf{W}$ is a Laplacian matrix (the proof is provided in Sec. A.2), which is a positive semidefinite matrix[b]. Therefore, the matrix $\bf{I} - \lambda \bf{W} = (1-\lambda) \bf{I} + \lambda (\bf{I}-\bf{W})$ is a positive definite matrix. Furthermore, due to the symmetry of the matrix, $(\bf{I} - \lambda \bf{W})$ is invertible.
> > >
> > >
> > > [a]Zhou D, Bousquet O, Lal T, et al. Learning with local and global consistency[J]. Advances in neural information processing systems, 2003, 16.
> > >
> > > [b] Russell M. Laplacian matrices of graphs: a survey. Linear algebra and its applications, 197:143–176, 1994

---

> > > > ### Comment · Reviewer_XKXi · 2023-11-22
> > > > **Thanks**
> > > >
> > > > Thanks for your clarification. Most of my concerns are addressed. Thus I would like to increase my score from 5 to 6.
> > > >
> > > > The major motivation of this paper is to improve the pseudo-labels. I would suggest thinking about whether there are some direct quantitive measures to indicate the quality of pseudo-labels. Currently, it focuses on exploring the improved quality of pseudo-labels by showing how it affects the final results. But the results are not very impressive (less than 1% improvement). It would be more convincing to directly show how the pseudo-labels are improved, e.g., improved accuracy or confidence of pseudo-labels. In addition, the computational efficiency can also be improved, e.g., the matrix inverse computation can have $\mathcal{O}(N^3)$, which is expensive when the batch size is large in real scenarios.
> > > >
> > > > Minor comments: Appendix A.2 uses $\lambda = \frac{1}{1+\mu}$, but section 3.3 uses $\lambda = \frac{\mu}{1+\mu}$. This inconsistency can be fixed in the revised version.

---

> > > > > ### Author Response · Authors · 2023-11-23
> > > > > **Thank You!**
> > > > >
> > > > > We are glad that most of your concerns are addressed. And we greatly appreciate that you will increase the score.
> > > > >
> > > > > Thanks for your valuable suggestions and constructive comments, which have further enhanced the quality of our paper. During the rebuttal, as suggested, we have included an additional experiment in Sec. D.2 and Table A.4 to explicitly evaluate the quality of pseudo-labels generated by PGM. We notice that the accuracy of pseudo-labels are improved. In our future work, we will further consider how to balance computational efficiency and accuracy. One promising direction could involve transforming the Laplacian matrix into a sparse matrix and employing a polynomial approximation method [c] to efficiently compute the matrix inversion. This approach could reduce the computation cost and improve the efficiency, which we will explore in future.
> > > > >
> > > > > Thank you for pointing out the typos in Sec 3.3. We have fixed it in our latest revision.
> > > > >
> > > > > [c]Smola A J, Kondor R. Kernels and regularization on graphs[C]//Learning Theory and Kernel Machines: 16th Annual Conference on Learning Theory and 7th Kernel Workshop, COLT/Kernel 2003, Washington, DC, USA, August 24-27, 2003. Proceedings. Berlin, Heidelberg: Springer Berlin Heidelberg, 2003: 144-158.

---

### Official Review · Reviewer_xjYH · 2023-10-31

**Soundness:** 3 good
**Presentation:** 2 fair
**Contribution:** 2 fair
**Rating:** 5
**Confidence:** 4

**Summary:**

This paper studies test-time adaptation method and propose a pseudo-label-based method, called PROGRAM. Authors utilize prototype graph construction and prototype graph based label propagation to obtain more accurate pseudo labels. A robust self-training strategy is proposed to employ cross entropy loss or symmetric cross entropy loss to all samples in the batch. Experiments across four domain generalization benchmarks and three corruption benchmarks are provided to evaluate the effectiveness of PROGRAM.

**Strengths:**

1. The experimental results are rich, which consists of four DG benchmarks and three corruption benchmarks.

2. The proposed method is clear, which is easy to follow.

**Weaknesses:**

1. The novelty of the paper remains to be discussed. Authors just use Prototype Graph Model directly and the strategy that selects some samples with CE loss and others with consistency regularization is commonly used in unsupervised learning. A similar paper [1] which also provides the derivation of Prototype Graph Model for few shot learning should be cited.

[1] Hao Zhu, and Piotr Koniusz. Transductive Few-shot Learning with Prototype-based Label Propagation by Iterative Graph Refinement. In Proc. CVPR, pages 23996--24006, 2023.

**Questions:**

1. The experiment of Fig.3 is weak, the class number of PACS is too small, authors should use benchmarks like CIFAR-100-C or OfficeHome to evaluate the sensitivity of batch size.

2. What is the accuracy and percentage of pseudo label after filtering by PGM?

Typo error: the caption of Table 4 : ↑ means higher is better.

---

> ### Author Response · Authors · 2023-11-21
> **Response to Reviewer xjYH (1/3)**
>
> **Q1-1: A similar paper (Zhu et al. 2023) [a] should be cited.**
>
> Thanks for bring this paper (Zhu et al. 2023) to our attention. We appreciate your suggestion and include a citation to this paper in our revision. However, we would like to highlight the distinguishing aspects between our work and Zhu et al.:
>
> 1. Different research topics and settings: Zhu et al. focus on transductive few-shot learning, where LABELED support samples and ALL unlabeled queries are available, assuming training and testing occur in the SAME domain. Our paper, on the other hand, addresses online test-time adaptation, which involves fine-tuning a pre-trained model using a BATCH of unlabeled test data from the TARGET domain, without accessing to ANY labeled data.
> 2. Distinct graph construction: In Zhu et al. (2023), prototypes are not part of the graph and are updated separately using gradient descent. In contrast, we treat prototypes as graph vertices in our approach.
> 3. Varied prototype definitions: Zhu et al. (2023) initializes prototypes as mean vectors of labeled support samples since labeled data is available. In contrast, we initialize prototypes using the model weights of the classification layer.
> 4. Offline vs. online approach: Zhu et al. (2023) requires access to all unlabeled queries and performs iterative prototype updates in an offline manner. In contrast, in our online test-time adaptation, the unlabeled test set arrives in an online manner. Our re-initialization design ensures that our prototypes stay up to date with our model and maintain their representation of global class characteristics.
>
> By highlighting these differences, we emphasize the unique contributions and focus of our work compared to Zhu et al. (2023).
>
> [a] Hao Zhu, and Piotr Koniusz. Transductive Few-shot Learning with Prototype-based Label Propagation by Iterative Graph Refinement. In Proc. CVPR, pages 23996--24006, 2023.
>
> **Q1-2: Novelty concerns.**
>
> The major contribution of this paper is to introduce a novel PROGRAM method specifically designed for test-time adaptation (TTA). PROGRAM addresses the issues of noisy pseudo-labels in the context of TTA by introducing the Prototype Graph Model (PGM) for pseudo-label generation and Robust Self-training (RST) for test-time fine-tuning.
>
> (1) Existing methods overlook either the global class prototype information, or the local neighboring features of test samples, leading to noisy pseudo-labels. To address this issue, we explore a novel scheme of using flexible graph representation to blend the benefits of both prototype-based and nearest-neighbor based pseudo-labeling.
>
> (2) We creatively combine pseudo-labeling (PL) and consistency regularization (CR) for self-training. We analyze the limitations of PL and CR separately. Based on the analysis, we propose RST to combine the benefits of hard pseudo-labels for accelerating model convergence and soft pseudo-labels for noise resistance.

---

> ### Author Response · Authors · 2023-11-21
> **Response to Reviewer xjYH (2/3)**
>
> **Q2: Authors should use benchmarks like CIFAR-100-C or OfficeHome to evaluate the effect of batch size.**
>
>
> Thank you for your valuable suggestion! We have incorporated CIFAR-100-C and OfficeHome benchmarks into our evaluation to assess the impact of batch size, as you recommended. The results are presented below.
>
> We have obtained similar conclusions to those depicted in Figure 3. Our method consistently outperforms other approaches across various batch sizes. Moreover, as the batch size increases, the superiority of our proposed PROGRAM becomes even more pronounced, highlighting the advantages of our approach.
>
> **Effect of batch sizes on OfficeHome dataset with ResNet-50 backbone. Average accuracy(%) is reported. The higher the better.**
> | Method            |  8      | 16      | 32 | 64 | 128   |
> |-------------------|-----------|-----------|------------|-----------------|--------|
> | ERM (ResNet-50)   | 67.13±0.99 |  67.13±0.99 |  67.13±0.99 |  67.13±0.99    |  67.13±0.99|
> | +Tent             |  62.10±1.44 | 64.84±1.86 |  66.29±0.77| 66.67±1.26      | 67.00±1.80  |
> | +SHOT             | 66.07±0.85 | 67.00±1.20 | 67.65±0.72 | 67.73±1.37      | 67.99±1.41  |
> | +PL               | 57.51±3.26 | 60.70±3.12 | 62.85±3.05 | 63.32±1.92      | 63.70±1.81  |
> | +T3A              | 68.60±0.90 | 68.55±1.23 | 68.26±0.84 | 68.35±0.72      | 68.47±1.26   |
> | +TAST             |  68.70±0.60 | 68.76±0.97 |  68.63±0.70 |  68.60±0.53     |  68.99±1.24  |
> | +TAST-BN          | 68.04±1.67 |68.42±1.12  | 68.88±0.50 |  69.23±1.55     |69.36±1.98  |
> | +TSD              |67.59±1.63  | 68.02±1.00 | 68.43±0.74 | 68.72±0.96      | 69.12±1.11  |
> | +PROGRAM          | **68.80±1.33** | **68.93±1.25** | **69.05±0.66** | **69.58±1.46** | **69.76±1.29**|
>
> **Effect of batch sizes on CIFAR-100C dataset with ResNet-50 backbone. Average error rate (\%) is reported. The lower the better.**
> | Method            |  8      | 16      | 32 | 64 | 128   |
> |-------------------|-----------|-----------|------------|-----------------|--------|
> | ERM (ResNet-50)   |   60.35 |  60.35 |  60.35 |  60.35     |  60.35  |
> | +Tent             |  45.56 | 43.25 | 41.54 | 40.16      | 39.04  |
> | +SHOT             | 47.48 | 45.13 | 43.71 |42.27      |41.54 |
> | +PL               | 46.18 | 44.78 | 42.13 | 41.48      | 40.06  |
> | +T3A              |59.89|59.03 | 58.75 | 58.32     | 58.28  |
> | +TAST             | 64.12 | 62.63 | 61.54 | 60.96      | 60.74  |
> | +TAST-BN          | 43.45 | 41.87 | 39.12 | 38.48      | 37.82  |
> | +TIPI             | 39.36 | 39.14| 38.86 | 38.65     | 38.33  |
> | +TSD              | 39.61 | 39.20 |38.74| 38.16      | 37.67  |
> | +PROGRAM          |**38.97** |**38.63** | **37.95** | **37.28** |**36.42**|

---

> ### Author Response · Authors · 2023-11-21
> **Response to Reviewer xjYH (3/3)**
>
> **Q3: What is the accuracy and percentage of pseudo label after filtering by PGM?**
>
> We appreciate the reviewer's important question regarding the quality of pseudo-labels generated by PGM. First of all, We would like to clarify that our method does not filter out pseudo-labels. Instead, we introduce the integration of pseudo-labeling and consistency regularization to alternately fine-tuning the model in the RST module, which is one of the main innovations of our approach. By doing so, our method fully utilizes all pseudo-labels without explicitly filtering them. More details about this aspect can be found in the Introduction and Section 3.4 of our paper.
>
> In response to the question, we also conducted an additional experiment to explicitly evaluate the accuracy of the PGM module. The results are shown below.
>
> Exp ID| Method            | VLCS      | PACS      | OfficeHome | TerraIncognita | Avg.   |
> |---|-------------------|-----------|-----------|------------|-----------------|--------|
> | #1 | ERM (ResNet-50)     |  76.71±0.50 | 83.21±1.14 | 67.13±0.99 | 45.93±1.34      | 68.25  |
> | #2 | ERM + PGM (before fine-tune)           |  77.36±0.61| 84.16±0.93 | 67.69±1.35 | 46.38±1.79      | 68.90  |
> | #3 | ERM + PROGRAM          |  78.17±0.92 | 90.78±0.40 | 69.05±0.66 | 47.84±1.23 | 71.46|
> | #4 | ERM + PGM (after fine-tune)          |  78.73±1.27 | 91.30±0.63 | 69.57±0.88 | 48.31±1.42 | 71.98|
>
> Previous TTA methods commonly utilize the pseudo-labels generated by ERM directly (#1). The accuracy of the generated pseudo-labels by PGM is reported in #2. By comparing results #1 and #2, we demonstrate that PGM can produce superior pseudo-labels compared to the original ERM (68.90 vs. 68.25). We then leverage these generated pseudo-labels for robust self-training, resulting in the improved performance of model #3 (71.46). In #4, we report the results obtained by applying PGM to generate the next round of pseudo-labels after fine-tuning. Consistent improvement is observed (71.98 vs. 71.46), highlighting the potential for iterative refinement. However, to ensure fair comparisons with other TTA methods, we do not report the results of #4 or the outcome of iterative refinement. These findings demonstrate that our method can produce more reliable pseudo-labels, leading to enhanced performance when compared with other TTA methods. We have included these results in the revised version.
>
> **Q4: Typo error.**
>
> Thank you very much for pointing it out. We have fixed it in the revised version.

---

> > ### Author Response · Authors · 2023-11-23
> > **Official Comment by Authors**
> >
> > Dear Reviewer xjYH,
> >
> > We have tried our best to address all the concerns and provided as much evidence as possible. May we know if our rebuttals answer all your questions? We truly appreciate it.
> >
> > Best,
> >
> > 7713 Authors

---

### Official Review · Reviewer_zHCF · 2023-10-31

**Soundness:** 2 fair
**Presentation:** 2 fair
**Contribution:** 3 good
**Rating:** 6
**Confidence:** 3

**Summary:**

This paper proposes a new test-time adaptation method, which has two contributions: 1) combines both the prototype-based pseudo label and nearest-neighbor-based pseudo label in a prototype graph to generate a comprehensive pseudo label and 2) Combines hard and soft pseudo-labels to improve the adaptation.

**Strengths:**

+The paper is well-written, and the overview figure clearly demonstrates the main contributions of this work.
+It is interesting to generate the pseudo-label by label propagating within the constructed graph.
+Experimental results are impressive, the proposed methods got SOTA in all proposed benchmarks.

**Weaknesses:**

Methodology:
-PGM is a non-parametric process to generate the pseudo-labels.
The motivation for using prototypes to determine the connectivity between two vertices is unclear. Could you use dot-product or cosine similarity to replace this? Some experimental comparisons and analysis could provide more insights.

-PGM lacks comparisons with non-parametric attention modules. Given the predefined graph G, the non-parametric attention module could first update each node in the graph with the help of the adjacency matrix, and then use the updated class prototypes to make predictions of each updated test feature.

-Could the author provide more insights into why PGM shows more reliable pseudo-labels?  Especially, what does “more reliable” mean?

-Some confusion about the derivations in A3 and A4. How to simplify A3 as A4. Could the author provide more detailed information?

**Questions:**

see weaknesses

---

> ### Author Response · Authors · 2023-11-21
> **Response to Reviewer zHCF (1/2)**
>
> **Q1-1: The motivation for using prototypes to determine the connectivity between two vertices is unclear.**
>
> Prototype-based graph construction is based on the idea that we can use a small number of prototypes to turn sample-to-sample affinity computations into much simpler sample-to-prototype affinity computations. In addition, as prototypes contain global class-level information, they provide more robust and stable representations. This helps to mitigate the impact of noisy test samples as well as class imbalance (please also refer to Q4 in the response to Reviewer C9fo for more details about class imbalance).
>
> **Q1-2: Could you use dot-product or cosine similarity to replace the affinity computation?**
>
> Thank you for your insightful suggestions. We have made the requested changes and conducted additional experiments to address your concerns. Specifically, we replace the affinity computation in our graph construction with dot-product, as you suggested. The experimental results are shown in the table below. All models use ResNet-50 as the backbone. These results also provide clear validation of the effectiveness of our proposed prototype-based graph construction and label propagation. The suboptimal performance of using the dot product is attributed to the fact that prototypes represented by linear layers and the test features may exist in different feature spaces and not follow the same distribution. Therefore, directly calculating the dot product is not an optimal approach. In contrast, employing the probability distribution method to construct the graph eliminates this issue.
>
> |Exp ID| Method            | VLCS      | PACS      | OfficeHome | TerraIncognita | Avg.   |
> |-----------------|-------------------|-----------|-----------|------------|-----------------|--------|
> |#1| ERM  (ResNet-50)  |  76.71±0.50 | 83.21±1.14 | 67.13±0.99 | 45.93±1.34      | 68.25  |
> |#2| ERM + PGM (dot-product) +RST   |  77.92±1.43 | 90.15±0.82 | 68.62±0.91 | 47.25±1.54      |  70.98|
> |#3| ERM + PROGRAM          |  **78.17±0.92** | **90.78±0.40** | **69.05±0.66** | **47.84±1.23** | **71.46**|
>
>
>
> **Q2: PGM lacks comparisons with non-parametric attention modules.**
>
> Thanks for the insightful suggestion. We have conducted additional experiments to compare PGM with non-parametric attention modules as suggested. The results are presented in the table below. The attention module have difficulty in efficiently exchanging message between prototypes represented by linear layers and the test features that may reside in different feature spaces and not follow the same distribution. So using the features from the test samples to update prototypes is not a good practice. By constructing graphs using probability distributions, we can avoid updating prototypes with test features. This method leads to better performance. These findings demonstrate that our PGM outperforms the attention module alternative (71.46 vs 70.73), confirming the effectiveness of our proposed prototype-based graph construction.
>
> |Exp ID| Method            | VLCS      | PACS      | OfficeHome | TerraIncognita | Avg.   |
> |-------|-------------------|-----------|-----------|------------|-----------------|--------|
> |#1| ERM (ResNet-50)     |  76.71±0.50 | 83.21±1.14 | 67.13±0.99 | 45.93±1.34      | 68.25  |
> |#2| ERM + attention module +RST   |  77.68±1.51 | 89.85±1.22 | 68.29±1.35 | 47.12±2.03      |  70.73 |
> |#3| ERM + PROGRAM          |  **78.17±0.92** | **90.78±0.40** | **69.05±0.66** | **47.84±1.23** | **71.46**|

---

> ### Author Response · Authors · 2023-11-21
> **Response to Reviewer zHCF (2/2)**
>
> **Q3: Could the author provide more insights into why PGM shows more reliable pseudo-labels? Especially, what does “more reliable” mean?**
>
> Thanks for the reviewer's question. The term "more reliable" refers to higher accuracy in this context. Please kindly note that the motivation behind the PGM module has been extensively discussed in our paper. We have provided a thorough analysis of both prototype-based pseudo-labeling and nearest-neighbor pseudo-labeling, highlighting their respective strengths and limitations. PGM combines the advantages of both approaches by incorporating global representative features of prototypes and local information from neighboring test data through a flexible graph representation.
>
> To further demonstrate the effectiveness of PGM in generating more reliable pseudo-labels, we conduct an additional experiment. The quality of pseudo-labels generated by PGM was explicitly evaluated in the following results.
>
>
> Exp ID| Method            | VLCS      | PACS      | OfficeHome | TerraIncognita | Avg.   |
> |---|-------------------|-----------|-----------|------------|-----------------|--------|
> | #1 | ERM (ResNet-50)     |  76.71±0.50 | 83.21±1.14 | 67.13±0.99 | 45.93±1.34      | 68.25  |
> | #2 | ERM + PGM (before fine-tune)           |  77.36±0.61| 84.16±0.93 | 67.69±1.35 | 46.38±1.79      | 68.90  |
> | #3 | ERM + PROGRAM          |  78.17±0.92 | 90.78±0.40 | 69.05±0.66 | 47.84±1.23 | 71.46|
> | #4 | ERM + PGM (after fine-tune)          |  78.73±1.27 | 91.30±0.63 | 69.57±0.88 | 48.31±1.42 | 71.98|
>
> Previous TTA methods commonly utilize the pseudo-labels generated by ERM directly (#1). The accuracy of the generated pseudo-labels by PGM is reported in #2. By comparing results #1 and #2, we demonstrate that PGM can produce superior pseudo-labels compared to the original ERM (68.90 vs. 68.25). We then leverage these generated pseudo-labels for robust self-training, resulting in the improved performance of model #3 (71.46). In #4, we report the results obtained by applying PGM to generate the next round of pseudo-labels after fine-tuning. Consistent improvement is observed (71.98 vs. 71.46), highlighting the potential for iterative refinement. However, to ensure fair comparisons with other TTA methods, we do not report the results of #4 or the outcome of iterative refinement. These findings demonstrate that our method can produce more reliable pseudo-labels, leading to enhanced performance when compared with other TTA methods. We have also included these results in our revised version for further clarification.
>
> **Q4: How to simplify A3 as A4?**
>
> We provide a more detailed explanation of the simplification process for this formula as follows.
>
> Starting from Equation A3:
> \begin{equation}
>  \mathbf{Y}^* - \mathbf{W}\mathbf{Y}^* + \mu (\mathbf{Y}^* - \mathbf{Z}) = 0,
> \end{equation}
>
> We expand the parentheses at the end to obtain the following expression:
>
> \begin{equation}
>  (1+\mu) \mathbf{Y}^* - \mathbf{W}\mathbf{Y}^* -  \mu \mathbf{Z} = 0,
> \end{equation}
>
> Dividing both sides of the equation by (1 + $\mu$), we get Equation A4:
>
> \begin{equation}
> \mathbf{Y}^* - \frac{1}{ 1+\mu }\mathbf{W} \mathbf{Y}^* -  \frac{\mu}{1+\mu} \mathbf{Z} = 0,
> \end{equation}
>
> We have also made modifications in the revised version to enhance the clarity of the paper.

---

> > ### Comment · Reviewer_zHCF · 2023-11-22
> > **Response to the authors**
> >
> > Thanks for the rebuttal. It solves most of my concerns. I will keep my score and suggest that the paper is above the acceptance threshold.

---

> > > ### Author Response · Authors · 2023-11-23
> > > **Thank you!**
> > >
> > > We are glad that most of your concerns are addressed. And we greatly appreciate your acknowledgement of our work. Thanks for your valuable suggestions and constructive comments, which have further enhanced the quality of our paper.

---

### Official Review · Reviewer_DVoa · 2023-11-01

**Soundness:** 3 good
**Presentation:** 2 fair
**Contribution:** 3 good
**Rating:** 6
**Confidence:** 2

**Summary:**

This paper proposes the PROGRAM, a TTA method that is comprised of the Prototype Graph Model (PGM) for pseudo-label generation and Robust Self-training (RST) in self-training. PGM is designed to blend the benefits of both prototype-based and nearest-neighbor based pseudo-labeling. RST combines pseudo-labeling and consistency regularization. Experimental results validate the efficacy of the PROGRAM method.

**Strengths:**

1.	PROGRAM looks like an interesting approach to combine the benefits of prototype-based and nearest-neighbor based pseudo-labeling.
2.	The paper presents extensive experiments to validate the effectiveness of the method.

**Weaknesses:**

The presentation of the paper could be improved:
* The explanation for Figure 1 appears to lack details. A clearer description would be helpful, such as specifying that the red line represents the decision boundary. Moreover, the meaning of the red dashed line in 1(b) is ambiguous.
* The meaning of the symbols ‘+’ used in Tables 1 and 2 is not clear. It would be helpful if their meaning can be clarified.

**Questions:**

1.	While PROGRAM's runtime is competitive as a whole feature extractor, it noticeably lags behind some partial ones like T3A. Given that the performance edge of PROGRAM over T3A isn't substantial in some tasks (as shown in Table 1), could you shed light on the specific scenarios where PROGRAM would be the preferred choice? Essentially, under what circumstances should one opt for PROGRAM, even at the expense of computational speed?

2.	Could you provide a breakdown of PROGRAM's runtime between PGM and RST (Table 6)?

3.	From Table 3, RST seems to have a small impact on improving results. Furthermore, ResNet-50 on its shows good results. How do you justify the improvements brought by PGM and RST given their relatively modest contribution to performance?

---

> ### Author Response · Authors · 2023-11-21
> **Response to Reviewer DVoa (1/2)**
>
> **Q1: Suggestions on improving the clarity of Figure 1.**
>
> Thank you for your valuable feedback. We have made the necessary refinements to Figure 1 to enhance its clarity and improve readability.
>
> **Q2: The meaning of the symbols ‘+’ used in Tables 1 and 2 is not clear.**
>
> We apologize for any confusion caused. The symbol '+' indicates the application of the TTA method to the ERM baseline method. Each row in Tables 1 and 2 represents the evaluation accuracy of the ERM baseline combined with the respective TTA method.
>
>
> **Q3: Could you shed light on the specific scenarios where PROGRAM would be the preferred choice compared with partial-update methods? Under what circumstances should one opt for PROGRAM, even at the expense of computational speed?**
>
> Our method, PROGRAM, excels in scenarios where there is a substantial domain shift between the source and target data. Partial-update methods, such as T3A and TAST, assume that the feature extractor backbone trained on the source domain performs well on the target domain, and only adjust the final linear classification layer. However, when the domain shift is significant, the assumption does not hold and may yield poor results without adapting the backbone.
>
> Our experimental results support this analysis. Li et al. [a] pointed out that the VLCS dataset has a relatively smaller domain shift compared to the PACS dataset. In Table 1 of our main text, we observe that PROGRAM demonstrates a more pronounced improvement on datasets with significant domain shifts, such as the PACS dataset (+7.2% compared to T3A), compared to datasets with smaller domain shifts like VLCS (+0.49% compared to T3A). Moreover, on image corruption benchmarks with substantial domain gaps, our approach significantly outperforms T3A (average improvement of 20.34%). While the improvement on datasets with smaller domain gaps may not be as evident, there is still a notable and consistent enhancement.
>
> As suggested by Reviewer XKXi, we also investigated a partial-update variant of PROGRAM, which only fine-tuned the linear classification layer (#5). Based on the experimental results below, fine-tuning only the linear layer (#5) achieves a good trade-off between accuracy (66.60%) and efficiency (13.05s). For instance, it achieves better accuracy than TAST (66.60% vs 66.39%), while being more efficient (13.05s vs 18.07s).
>
>
> | Exp ID | Method                                       | VLCS       | PACS       | OfficeHome | TerraIncognita | Avg.  | Runtime (s) |
> | ------ | -------------------------------------------- | ---------- | ---------- | ---------- | -------------- | ----- | ----------- |
> | #1     | ERM (ResNet-18)                              | 74.88±0.46| 79.29±0.77| 62.10±0.31 |40.62±1.19| 64.22|11.15|
> | #2     | +T3A (Iwasawa & Matsuc, 2021)   | 77.26±1.49| 80.83±0.67 |63.21±0.50| 40.20±0.60| 65.38|11.73|
> | #3     | +TAST (Jang et al. 2023)       | 77.27±0.67 |81.94±0.44 |63.70±0.52| 42.64±0.72|66.39|18.07|
> | #4     | +TAST-BN (Jang et al. 2023)       |  75.21±2.36| 87.07±0.53| 62.79±0.41| 39.43±2.24| 66.13 |85.08|
> | #5     | +PROGRAM (fine-tune linear)                 | 77.52±0.86 | 82.27±1.09 | 63.88±1.00 | 42.73±1.75    | 66.60 | 13.05       |
> | #6     | +PROGRAM (our final version: fine-tune all) | 77.75±1.37| 88.03±0.74| 64.59±0.85 |43.16±1.12| 68.38|29.07|
>
>
>
> [a] Li, Da, et al. "Deeper, broader and artier domain generalization." Proceedings of the IEEE international conference on computer vision. 2017.

---

> ### Author Response · Authors · 2023-11-21
> **Response to Reviewer DVoa (2/2)**
>
> **Q4: Could you provide a breakdown of PROGRAM's runtime?**
>
> Thanks for the advice. We report the breakdown of PROGRAM's runtime below.
>
> **Absolute runtime performance analysis with ResNet-18 backbone on a single Titan XP GPU on the PACS dataset.**
> | Exp ID | Method                           | Runtime (s)           |
> | ------ | -------------------------------- | --------------------- |
> | Baseline  |                             |                       |
> | #1     | ERM (ResNet-18)                  | 11.15                 |
> | #2     | ERM + SHOT (Liang et al, 2020)   | 11.15 + 20.97 = 32.12 |
> | #3     | ERM + SHOTIM (Liang et al, 2020) | 11.15 + 20.73 = 31.88 |
> | #4     | ERM + TSD (Wang et al, 2023)     | 11.15 + 19.73 = 30.88 |
> | Ours   |                                  |                       |
> | #5     | ERM + PGM                        | 11.15 + 1.21 = 12.36  |
> | #6     | ERM + RST                        | 11.15 + 16.78 = 27.93 |
> | #7     | **ERM + PGM + RST (PROGRAM)**   | 11.15 + 17.92 = 29.07 |
> | #8     | ERM + PGM + CE Loss              | 11.15 + 17.54 = 28.69 |
> | #9     | ERM + PGM + KL Loss              | 11.15 + 17.65 = 28.80 |
>
>
> Our PGM module exhibits a relatively short computational time (comparing #1 and #5), making it efficient for reliable pseudo-label generation. And the main time consumption occurs during the back-propagation phase in RST. However, the computational time required for the RST module is comparable to other self-training modules that employ different loss functions (compare #7 with #8 and #9). Overall, our method demonstrates smaller computational costs compared to TTA methods such as SHOT (#2), SHOTIM (#3), and TSD (#4) that involve adjusting the whole model parameters. This highlights the practicality and feasibility of incorporating our method into real-world applications, as it achieves significant performance gains while maintaining reasonable computational efficiency.
>
>
>
> **Q5-1: From Table 3, RST seems to have a small impact on improving results. How do you justify the improvements brought by PGM and RST given their relatively modest contribution to performance?**
>
> Please kindly note that in Table 1, with ResNet-50 as the backbone, TSD (Wang et al., 2023) improves upon TAST (Jang et al., 2023) by 0.43%, and TAST improves upon T3A (Iwasawa & Matsuc, 2021) by 0.69%. These improvements are considered substantial in the field of test-time adaptation (TTA). In comparison, RST alone brings an improvement of 0.63%. Moreover, when combined with PGM, our method (PROGRAM) achieves a more significant improvement over the previous state-of-the-art TSD by 1.57%. We believe that the performance boost achieved by PROGRAM is substantial and demonstrates its effectiveness in advancing the state-of-the-art in TTA.
>
>
> **Q5-2: ResNet-50 already shows good results.**
> While ResNet-50 may achieve good results in scenarios with small domain gaps, test-time adaptation (TTA) becomes crucial when dealing with large domain gaps. For instance, the PACS dataset exhibits a strong domain gap. Our approach shows a significant improvement over the ResNet-50 baseline (90.78% vs 83.21%) on PACS. Particularly, on image corruption datasets, our approach demonstrates substantial enhancement, with an average improvement of 23.91%. These results validate the necessity and effectiveness of our approach in addressing challenging scenarios with significant domain gaps.

---

> > ### Author Response · Authors · 2023-11-23
> > **Official Comment by Authors**
> >
> > Dear Reviewer DVoa,
> >
> > We have tried our best to address all the concerns and provided as much evidence as possible. May we know if our rebuttals answer all your questions? We truly appreciate it.
> >
> > Best,
> >
> > 7713 Authors

---

### Official Review · Reviewer_C9fo · 2023-11-03

**Soundness:** 3 good
**Presentation:** 3 good
**Contribution:** 3 good
**Rating:** 8
**Confidence:** 3

**Summary:**

This paper introduces a novel Test-Time Adaptation method named PROGRAM, which leverages a pseudo-labeling approach to enhance model performance for image corruption benchmarks. PROGRAM is build around two key components: (1) a Prototype Graph Model for generating pseudo-labels and (2) Robust Self-Training to adapt the model at test time. The proposed new method has been validated across various architectures, demonstrating consistent improvements over the existing TTA methods. Extensiv experiments show that PROGRAM not only outperforms other state-of-the-art methods but also maintains its performance across a range of hyperparameters. Additionally, it is designed to be "plug-and-play", easily integrating with different networks and TTA methods.

**Strengths:**

The PROGRAM demonstrates consistent performance improvements across a variety of backbone architectures. This versatility suggests that the method is robust and can be applied to a broad set of existing models, and across the range of hyperparameters.

Empirical validation: The method has been empirically validated with extensive experiments showing that it outperforms existing state-of-the-art methods on various domain generalization and image corruption benchmarks.

**Weaknesses:**

The paper does not discuss potential limitations or failure modes of the proposed method. While the paper reports on the efficiency of PROGRAM compared to other methods, there is limited discussion on the absolute runtime performance, especially in comparison to the baseline models without TTA.

Practical Integration Challenges: Despite claiming that PROGRAM is a "plug-and-play" solution, the paper does not delve into the practical challenges of integrating the method into different systems or architectures.

**Questions:**

Given the PROGRAM TTA approach does nto apply any modifications during training phase, can the effectiveness of PGM and RST in PROGRAM compensate for potential deficiencies in the pre-trained model? I.e. have you checked how the test time adaptation method works when the pre-trained model is suboptimal.

How does the graph construction technique manages the class imbalance that might be present in the unlabeled target data? Related to the discussion in section 3.2 about initialization of prototypes and constructing a prototype graph.

---

> ### Author Response · Authors · 2023-11-21
> **Response to Reviewer C9fo (1/3)**
>
> **Q1-1: Discuss potential limitations.**
>
> 1. It is still under explarion that how to extend our proposed PROGRAM to widespread computer vision tasks. Similar to the previous TTA methods, PROGRAM is currently applied to classification task and not adapted to regression problems such as object detection and semantic segmentation. We are actively working on creating a unified TTA framework that can be applied across all vision tasks.
>
> 2. Another problem is that it is hard to evaluate or estimate the model's behaviour in real application scenarios because the model is constantly updated with the test data. This may raise some ethical concerns in some sensitive applications. It's worth noting that the existing TTA methods, such as T3A, TAST, Tent, and TSD, also encounter similar challenges.
>
> **Q1-2: Discussion on the absolute runtime performance, especially in comparison to the baseline models without TTA.**
>
> Table 6 in our main text reports the extra runtime required by different TTA methods. As suggested, we also present the absolute runtime performance analysis below.
>
> **Absolute runtime performance analysis with ResNet-18 backbone on a single Titan XP GPU on the PACS dataset.**
> | Exp ID | Method                           | Runtime (s)           |
> | ------ | -------------------------------- | --------------------- |
> | Baseline  |                             |                       |
> | #1     | ERM (ResNet-18)                  | 11.15                 |
> | #2     | ERM + SHOT (Liang et al, 2020)   | 11.15 + 20.97 = 32.12 |
> | #3     | ERM + SHOTIM (Liang et al, 2020) | 11.15 + 20.73 = 31.88 |
> | #4     | ERM + TSD (Wang et al, 2023)     | 11.15 + 19.73 = 30.88 |
> | Ours   |                                  |                       |
> | #5     | ERM + PGM                        | 11.15 + 1.21 = 12.36  |
> | #6     | ERM + RST                        | 11.15 + 16.78 = 27.93 |
> | #7     | **ERM + PGM + RST (PROGRAM)**   | 11.15 + 17.92 = 29.07 |
> | #8     | ERM + PGM + CE Loss              | 11.15 + 17.54 = 28.69 |
> | #9     | ERM + PGM + KL Loss              | 11.15 + 17.65 = 28.80 |
>
> Our PGM module exhibits a relatively short computational time (comparing #1 and #5), making it efficient for reliable pseudo-label generation. And the main time consumption occurs during the back-propagation phase in RST. However, the computational time required for the RST module is comparable to other self-training modules that employ different loss functions (compare #7 with #8 and #9). Overall, our method demonstrates smaller computational costs compared to TTA methods such as SHOT (#2), SHOTIM (#3), and TSD (#4) that involve adjusting the whole model parameters. This highlights the practicality and feasibility of incorporating our method into real-world applications, as it achieves significant performance gains while maintaining reasonable computational efficiency.
>
>
> **Q2: Discussion on practical integration challenges.**
>
> Our PGM module is compatible with most pseudo-labeling based TTA methods (as shown in Table 4) and various network architectures (as shown in Table 5). We provide detailed instructions on integrating our method with other pseudo-labeling TTA methods in Appendix G.
>
> As regards the practical integration challenges, those TTA methods that do not use pseudo-labels may not be easily adapted because our method relies on pseudo-labeling. For example, techniques based on adjusting the statistical value of Batch Normalization (BN) layer may not be applicable to our proposed approach.

---

> ### Author Response · Authors · 2023-11-21
> **Response to Reviewer C9fo (2/3)**
>
> **Q3: How do test time adaptation methods work when the pre-trained model is suboptimal?**
>
> We appreciate the reviewer's valuable insight and we have added experiments to address this concern. To evaluate the effectiveness of our method with a suboptimal pre-trained model, we pre-train the ResNet-50 based source model on the source data with reduced training epochs (20 instead of 200) and modify learning rate (1e-5 instead of 5e-5). We then apply different TTA methods on top of this suboptimal model. The results are presented in the table below.
>
>
> | Method            | VLCS | PACS  | OfficeHome | TerraIncognita | Avg.   |
> |-------------------|-----------|-----------|------------|-----------------|--------|
> | ERM (ResNet-50, under-tuned) | 64.74±1.34 | 74.81±1.26 | 53.93±0.40 | 30.27±0.71      | 55.94 |
> | +Tent                        | 60.62±1.10 | 76.03±1.32 | 50.47±0.58 | 23.69±1.00      | 52.70 |
> | +TentAdapter                 | 57.15±1.76 | 75.34±1.74 | 53.58±1.43 | 29.70±1.89      | 53.94  |
> | +TentClf                     | 62.36±0.62 | 73.48±1.31 | 51.10±0.66 | 28.61±1.59      |53.89  |
> | +SHOT                        | 54.51±1.30 | 75.96±1.91 | 54.68±0.85 | 24.95±1.39      | 52.52 |
> | +SHOTIM                      | 53.28±1.96 | 75.75±1.47 | 54.49±1.15 | 24.30±1.54      | 51.96  |
> | +PL                          | 52.37±1.57 | 70.49±1.30 | 48.82±2.76 | 27.54±1.59      | 49.80  |
> | +PLCls                       | 64.79±1.44 | 73.43±1.54 | 51.80±1.60 | 31.57±1.42      | 55.40  |
> | +T3A                         | 65.64±0.41 | 75.80±1.09 | 54.91±1.50 | 30.90±0.96      | 56.81  |
> | +TAST                        | 66.56±0.82 | 76.52±1.68 | 55.04±1.11 | 31.38±1.54      | 57.37  |
> | +TAST-BN                     | 65.95±2.37 | 78.91±1.82 | 54.80±1.38 | 28.58±1.22      |57.06  |
> | +TSD                         | 64.38±2.09 | 78.67±2.01 | 55.02±1.67 | 31.86±2.12      | 57.48  |
> | +PROGRAM                     | **67.53±1.31** | **80.89±1.55** | **55.83±1.48** | **32.62±1.32** | **59.22**|
>
> From the table, we observe that when the pre-trained model is suboptimal, several approaches (e.g., Tent, SHOT and PL) fail to consistently improve over the baseline ERM. This is due to increased noise of pseudo-labels, which hampers the effectiveness of test-time adaptation. However, our PROGRAM consistently outperforms the baseline on all benchmarks, surpassing all competing TTA methods. This demonstrates that our method is robust and effective, even when working with a suboptimal pre-trained model. Our proposed PGM is able to generate reliable pseudo-labels, while RST facilitates test-time adaptation with noisy pseudo-labels. The experiment provides further validation of the efficacy of our proposed approach.

---

> ### Author Response · Authors · 2023-11-21
> **Response to Reviewer C9fo (3/3)**
>
> **Q4: How does the graph construction technique manage class imbalance in the unlabeled target data?**
>
> Our graph construction technique in PROGRAM considers the relation between prototypes and samples, rather than just the relation between samples. The prototypes provide more robust and stable representations of each class, which helps mitigate the impact of class imbalance in the test sample category distribution, enabling our method to produce high-quality pseudo-labels. It is important to note that our method does not assume a uniform class distribution prior and effectively handles class imbalance.
>
> In fact, we have conducted experiments on datasets with class imbalance, such as the PACS dataset. The table below shows that the class distribution of PACS is imbalanced. As shown in Table 1 of the main text, our approach achieves state-of-the-art performance on the PACS dataset.
>
>
> | Domain|Dog|Elephant|Giraffe|Guitar|Horse|House|Person|
> |-------------------|------------|-----------|-----------|------------|-----------------|--------|--------|
> |  A           | 379 | 255 | 285 | 184 | 201      | 295  |449|
> | C         | 389 | 457 | 346 | 135 | 324      | 288  | 405|
> | P          | 189 | 202 | 182 | 186 | 199      | 280  | 432|
> | S         | 772 | 740 | 753 | 608 | 816      |80  |160|
>
>
>
> To further assess the efficacy of our method in scenarios with severe class imbalance, we intentionally create a class-imbalanced dataset by reducing the number of images from specific classes in the PACS dataset. Specifically, we reduce the number of images in the "dog" and "giraffe" classes to one-fifth of their original count. We subsequently evaluated the performance of different test-time adaptation (TTA) methods on this imbalanced dataset. All methods are based on the ResNet-50 backbone. The results, summarized in the table below, demonstrate substantial improvements achieved by our method compared to other competing TTA methods, particularly in the presence of class imbalance.
>
> | Method            | PACS* (more imbalanced version)|
> |-------------------|-----|
> | ERM (ResNet-50)     | 82.73±1.36|
> | +Tent               | 84.01±0.57  |
> | +TentAdapter        | 82.54±1.49  |
> | +TentCls            | 81.78±0.95  |
> | +SHOT               |   82.97±1.04   |
> | +SHOTIM             | 83.09±1.58 |
> | +PL                 | 80.10±2.92  |
> | +PLCls              | 82.52±1.30  |
> | +T3A                | 83.07±0.42 |
> | +TAST               | 83.35±1.03  |
> | +TAST-BN            | 88.22±1.85  |
> | +TSD                | 88.36±1.28 |
> | +PROGRAM            |**90.20±0.61**|

---

> > ### Author Response · Authors · 2023-11-23
> > **Official Comment by Authors**
> >
> > Dear Reviewer C9fo,
> >
> > Thanks for your acknowledgement of our paper! We have tried our best to address all the concerns and provided as much evidence as possible. May we know if our rebuttals answer all your questions? We truly appreciate it.
> >
> > Best,
> >
> > 7713 Authors

---

### Author Response · Authors · 2023-11-21
**Response to All Reviewers**

We thank all reviewers for the constructive comments, which would further improve the quality of this paper.

We appreciate their findings that our method is interesting (Reviewer DVoa, zHCF), novel (Reviewer C9fo, XKXi) and versatile (Reviewer C9fo); our experiments are extensive (Reviewer C9fo, DVoa, xjYH) with impressive results (Reviewer zHCF, XKXi); and our presentation is clear and easy to follow (Reviewer zHCF, xjYH, XKXi).

All the modifications and additions in the main text and Appendix are highlighted in the revised PDF using red color. And we also individually respond to all the reviewers in the discussion boxes.

---

### Meta-Review · Area_Chair_hjYt · 2023-12-04

**Metareview:**

This paper considers the problem of test-time adaptation. To solve this problem, the authors propose a approach named PROGRAM, which is built with two components, Prototype Graph Model and Robust Self-Training. Experiments on various architectures and datasets validate the effectiveness over the existing methods. In addition, it can be integrated with different methods and achieves further improvements.


**Strengths**

- It is somewhat interesting to combine the benefit of prototype-based and nearest-neighbor based pseudo-labeling.

- The proposed method is robust to different architectures and methods. Good performance is achieved on various datasets.

- The paper is well-written, and the overview figure clearly demonstrates the main contributions of this work.

**Weaknesses**

- Potential limitations and failure modes are not well discussed.

- Novelty is somewhat limited, when compared to [A]. The techniques in the proposed method are explored in previous works (although in different tasks).

- High computational complexity.

- The absence of a thorough empirical evaluation regarding the quality of pseudo-labels.

[A] Hao Zhu, and Piotr Koniusz. Transductive Few-shot Learning with Prototype-based Label Propagation by Iterative Graph Refinement. In Proc. CVPR, pages 23996--24006, 2023.

**Justification For Why Not Higher Score:**

Potential limitations and failure modes are not well discussed; Novelty is somewhat limited compared to previous methods; High computational complexity; Lack of thorough empirical evaluation.

**Justification For Why Not Lower Score:**

This paper propose an interesting method to solve the TTA problem, which is robust to different architectures and methods and achieves state-of-the-art performance.

---

### Decision · Program_Chairs · 2024-01-16

Accept (poster)